# *Cryptovaranoides* is not a squamate

**Michael W Caldwell**[1,2]*, **Chase D Brownstein**[3,4], **Dalton L Meyer**[5], **Simon G Scarpetta**[6,7], **Michael SY Lee**[8,9], **Tiago R Simões**[10]

[1]Department of Biological Sciences, University of Alberta, Edmonton, Canada; [2]Department of Earth and Atmospheric Sciences, University of Alberta, Edmonton, Canada; [3]Department of Ecology and Evolutionary Biology, Yale University, New Haven, United States; [4]Stamford Museum and Nature Center, Stamford, United States; [5]Department of Earth and Planetary Sciences, Yale University, New Haven, United States; [6]Museum of Vertebrate Zoology, Department of Integrative Biology, University of California, Berkeley, Berkeley, United States; [7]Department of Environmental Science, University of San Francisco, San Francisco, United States; [8]College of Science and Engineering, Flinders University, Adelaide, Australia; [9]Earth Sciences Section, South Australian Museum, Adelaide, Australia; [10]Department of Ecology and Evolutionary Biology, Princeton University, Princeton, United States

*For correspondence:
mw.caldwell@ualberta.ca

Competing interest: The authors declare that no competing interests exist.

## eLife Assessment

Cryptovaranoides, a Late Triassic animal (some 230 Ma old), was originally described as a possibly anguimorph squamate, i.e., more closely related to snakes and some extant lizards than to other extant lizards, making Squamata much older than previously thought and providing a new calibration date inside it. Following a rebuttal and a defense, this fourth **important** contribution to the debate makes a **convincing** argument that Cryptovaranoides is not a squamate. Further comparisons to potentially closely related animals such as early lepidosauromorphs would greatly benefit this study, and parts of the text require clarification.

**Abstract** Accurate reconstruction of the timescale of organismal evolution requires placement of extinct representatives among living branches. In this way, the fossil record has the capacity to revise hypotheses of organismal evolution by producing representatives of clades that far predate the age of the clade inferred using phylogenies built from molecular data and previous fossil calibrations. Recently, one fossil with the potential to drastically change current understanding surrounding the timescale of reptile diversification was described from Triassic fissure-fill deposits in the United Kingdom. This taxon, †*Cryptovaranoides microlanius*, was originally placed deep within the squamate crown clade, suggesting that many lineages of living lizards and snakes must have appeared by the Triassic and implying long ghost lineages that paleontologists and molecular phylogeneticists have failed to detect using all other available data. Our team challenged this identification and instead suggested †*Cryptovaranoides* had unclear affinities to living reptiles, but a crown-squamate interpretation was later re-iterated by the team that originally described this species. Here, we again challenge the morphological character codings used to support a crown squamate affinity for †*Cryptovaranoides microlanius* and illustrate several empirical problems with analyses that find this taxon is a crown squamate. Our analyses emphasize the importance of stringency in constructing hypodigms of fossils, particularly when they may be key for proper time calibration of the Tree of Life.

## Introduction

Paleontology has found an important role in the era of widespread genome sequencing of living phylogenetic diversity: providing justification for the placement of fossil calibrations along molecular phylogenies (*Inoue et al., 2010*; *Heath et al., 2014*; *Near, 2004*; *Near et al., 2005*; *Parham et al., 2012*; *Near and Sanderson, 2004*). Placing fossils on the Tree of Life is common practice, and new discoveries (*Field et al., 2020*; *Simões et al., 2018*; *Brownstein, 2023*; *Velazco et al., 2022*) and reshuffling of hypothesized phylogenetic relationships (*Budd and Mann, 2024*) are constantly revising what fossils are best to use as prior calibration constraints. These factors make the robust placement of key fossil taxa essential for properly calibrating phylogenies in time, which themselves form a foundation for modern evolutionary biology (*Smith et al., 2020*).

Recently, *Whiteside et al., 2022* described †*Cryptovaranoides microlanius* based on a partially articulated skeleton and a collection of referred material from the Carnian (*Simms and Drost, 2024*) to Norian-Rhaetian (*Whiteside et al., 2022*; *Walkden et al., 2021*; *Chambi-Trowell et al., 2019*; *Butler et al., 2024*; 237–201.5 million years ago) fissure fill deposits of England, UK. In a paper (*Brownstein et al., 2023*) published at the end of 2023, we joined to refute the affinities of †*C. microlanius* to Anguimorpha, a deeply nested crown squamate clade, proposed by *Whiteside et al., 2022*. In their response to our paper, *Whiteside et al., 2024* disagree with many of our anatomical observations and restate their position on the affinities of †*C. microlanius*. *Whiteside et al., 2024* referred additional Late Triassic fossils to †*Cryptovaranoides microlanius* in support of *Whiteside et al., 2022* and presented phylogenetic results that this taxon is a crown group squamate (squamate hereafter). *Whiteside et al., 2024* is also an impassioned rebuttal to *Brownstein et al., 2023*, who substantially revised the description of †*Cryptovaranoides*, and upon correction of numerous character scorings in *Whiteside et al., 2022*, found this taxon to be '*either an archosauromorph or an indeterminate neodiapsid*' and not a lepidosaur, much less a crown squamate.

Here, we provide point-by-point refutations of the interpretations of both papers of *Whiteside et al., 2022*; *Whiteside et al., 2024* and describe what we consider to be major methodological errors in the comparative anatomical work and phylogenetic analyses they conduct. We also emphasize that both (*Whiteside et al., 2022*; *Whiteside et al., 2024*) papers fail to replicate the inferred position of †*C. microlanius* across phylogenetic analyses of the same and across different morphological datasets. We also highlight here where (*Whiteside et al., 2024*) should have reported their recovered synapomorphies; they elected instead to report results from other studies as though they had been recovered in the analyses conducted by *Whiteside et al., 2024*.

## Results

### Factual errors in *Whiteside et al., 2024*

*Whiteside et al., 2022*; *Whiteside et al., 2024* included numerous substantive comparative anatomy errors that can broadly be grouped into two categories: (i) anatomical interpretations, and (ii) translating these interpretations into scorings in different phylogenetic datasets. In *Brownstein et al., 2023*, we identified 22 errors in the study of *Whiteside et al., 2022* of type (i) ('Results' in *Brownstein et al., 2023*) and several more of type (ii) ('Supplementary Material' in *Brownstein et al., 2023*). In turn, *Whiteside et al., 2024* claimed that there were five observational errors in *Brownstein et al., 2023*. Four of these are supposed observational errors (type (i)) and one concerned a difference in how *Brownstein et al., 2023* and *Whiteside et al., 2024* score a character (type (ii)). This implies that *Whiteside et al., 2024* admitted that the other 18 errors type (i) produced by their previous study were correctly identified by *Brownstein et al., 2023*. Yet, *Whiteside et al., 2024* discussed additional characters in sections of their study and provided different interpretations of the anatomy of †*Cryptovaranoides microlanius* than did (*Brownstein et al., 2023*), but without clear justification. Here we focus on the four supposed observational errors (type (i)) listed by *Whiteside et al., 2024*.

### Entepicondylar and ectepicondylar foramina of humerus

*Whiteside et al., 2024* provided additional photographs of features on the distal ends of the humeri referred to †*Cryptovaranoides microlanius* that they maintained are homologous with the entepicondylar and ectepicondylar foramina observed in some, but not all, squamates. We disagree about the

identity of the features that *Whiteside et al., 2024* identified as these paired distal foramina for the following reasons:

1. First, the features on the humeri that *Whiteside et al., 2024* figured are not foramina, but fossae on the posterodistal surface of the humerus that are filled in with sediment. Although *Whiteside et al., 2024* claimed to observe this in a third isolated and referred humerus (NHMUK R38929), they neither figured this third humeral fragment, nor did they figure the internal structure of any of the supposed humeral heads, nor the evidence for referring these isolated elements to †*C. microlanius*. Contrary to *Whiteside et al., 2024*, computed tomography scans provide essential information about the structure of these fossae and show that, when infill is removed, foramina are absent (Figure 4 in *Brownstein et al., 2023*).

2. The structures that *Whiteside et al., 2024* interpreted to be the entepicondylar and ectepicondylar foramina on the holotype and referred humeri are also in the wrong place on the bone to be homologized as such. In all crown reptiles that possess these foramina, they are placed low on the anterior surface (entepicondylar) and high on the distal surface (ectepicondylar) of the humerus (e.g. see *Simões et al., 2018*; *Gauthier et al., 2012* for examples in squamates, *Hermanson et al., 2024* for examples in turtles, and finally *Simões et al., 2022b*; *DeMar et al., 2022* for examples in rhynchocephalians) and are dissimilar in shape and size (the entepicondylar foramen is elongated along the long axis of the humerus; the ectepicondylar foramen is circular). The features that *Whiteside et al., 2024* figured on the holotype and referred humeri of †*C. microlanius* are very similar in shape and size, are oddly both placed on the same side of the bone, and differ in placement from the foramina observed in other exemplar fossils and living species of reptiles (*Simões et al., 2018*; *Gauthier et al., 2012*; *Hermanson et al., 2024*; *Simões et al., 2022b*; *DeMar et al., 2022*). The morphology of the fossae observed on the distal end of the humeri referred to †*C. microlanius* by *Whiteside et al., 2024* is, however, similar in placement, size, and shape to the fossae described on the distal ends of the humeri of some archosauromorphs, including the azendohsaurid †*Puercosuchus traverorum* (Figure 13a in *Marsh et al., 2022*).

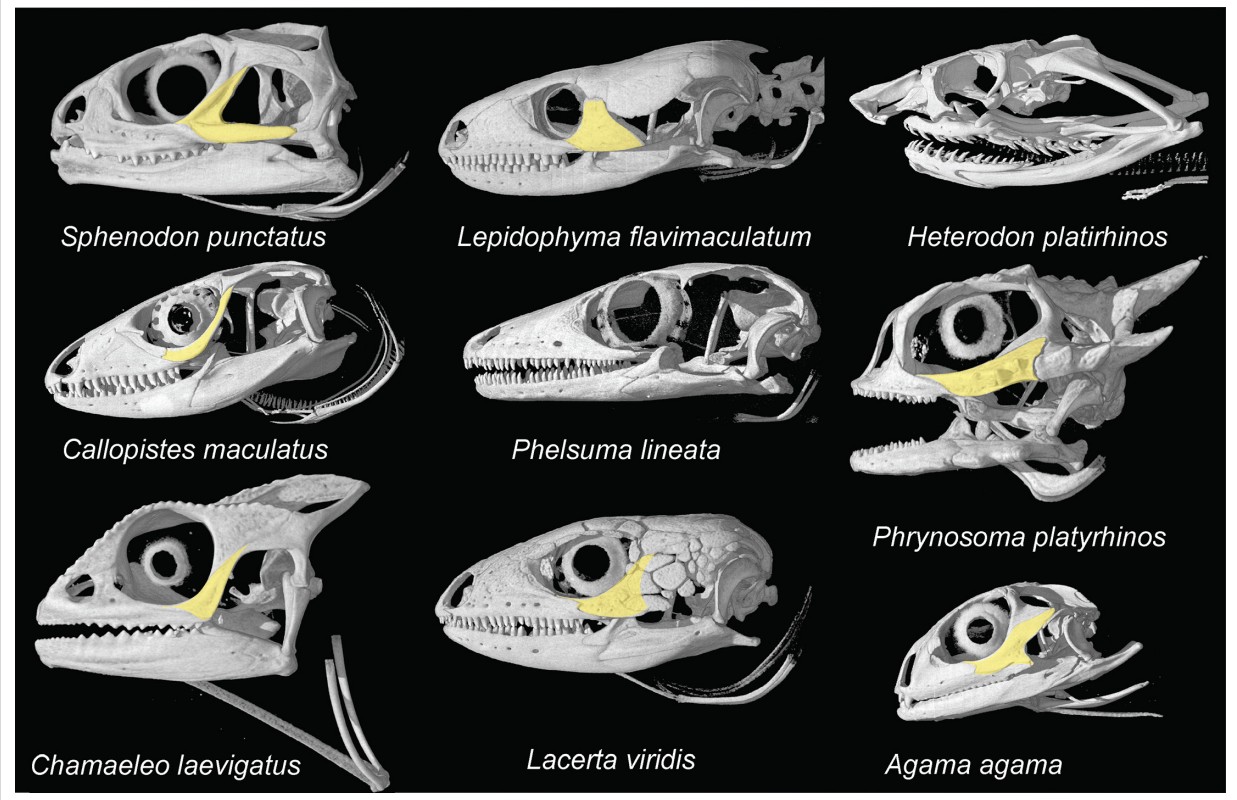

**Figure 1.** Comparison of jugal morphologies among living lepidosaurs. Note the variability in the presence and development of the posterior process, as well as the presence of the jugal itself. CT scan images are from digimorph.org.

3. We reiterate that the presence of these foramina is not an unambiguous synapomorphy of crown Squamata and is present across living reptilian diversity (*Simões et al., 2018*; *Gauthier et al., 2012*; *Simões et al., 2022b*). We note that, even if their interpretation is accurate, although the ectepicondylar foramen is variably present across squamates, the entepicondylar foramen is always absent in all crown squamates (*Simões et al., 2018*; *Gauthier et al., 2012*). Among lepidosaurs, the presence of both foramina would be found only among sphenodontians and stem lepidosaurs, such as †*Palaeagama* (*Simões et al., 2018*; *Simões et al., 2020*). Further, these foramina are also present in many non-lepidosaurian reptiles, including captorhinids (e.g. †*Captorhinus*), younginiforms (e.g. †*Hovasaurus* and †*Youngina*), *Claudiosaurus*, Acleistorhinidae (*Delorhynchus*), *Mesosaurus*, the possible stem turtle †*Eunotosaurus africanus*, and in some sauropterygians (e.g. *Serpianosaurus* and *Lariosaurus*; *Simões et al., 2018*; *Hill, 2005*; *Simões et al., 2022a*). In conclusion, if anything, the presence of both foramina would support the assignment of †*Cryptovaranoides* as outside of crown Squamata.

## Absence of jugal posterior process

*Whiteside et al., 2024* stated: '*Except for a very few fossil taxa, including two polyglyphanodontians,* †Tianyusaurus *and* †Polyglyphanodon, … *squamates lack a posterior process on the jugal.*' This is entirely incorrect. The two extinct taxa noted by *Whiteside et al., 2024* here are only unusual among squamates in the fact that they have a complete lower temporal bar, and thus an elongated jugal posterior process (*Gauthier et al., 2012*; *Simões et al., 2016*; *Keqin and Norell, 2000*; *Mo et al., 2010*). However, most families of squamates include species with a jugal posterior process (*Figure 1*; *Simões et al., 2018*; *Gauthier et al., 2012*; *Keqin and Norell, 2000*; *Estes et al., 1988*; *Brownstein et al., 2022*), even if a complete lower temporal bar is not present. Thus, the presence of a jugal posterior process was incorrectly scored in the datasets used by *Whiteside et al., 2024* (type (ii) error).

## Anterior emargination of the maxillary nasal process

*Whiteside et al., 2024* stated that (*Brownstein et al., 2023*) "…*seemingly relied on the anteriorly broken left maxilla*" and "*The anterior margin of the maxillary nasal process of the right maxilla tapers anteriorly…with no evidence of the type of emargination suggested*". The character in question (ch. 18 in the dataset used by *Whiteside et al., 2024*) is one of the original characters from *Simões et al., 2018*, which is the basis for the dataset *Tałanda et al., 2022* used by *Whiteside et al., 2024*. In both datasets, this character is described as '*Maxillae,* **posterior emargination**, *between nasal and orbital processes*'. [boldface added], and this character is extensively described in *Simões et al., 2018*. Therefore, *Whiteside et al., 2024* incorrectly assessed the anterior margin of the maxilla instead of the posterior margin.

## Expanded radial condyle of the humerus

*Whiteside et al., 2024* (p. 3) stated that (*Whiteside et al., 2022*) '…*noted an expanded radial condyle on the humerus*'. *Whiteside et al., 2024* (p.3, *Figure 1b*) wrote in their figure caption regarding another isolated fragment: '*(b) NHMUK PV 38911 isolated larger specimen of the distal end of left humerus of* †Cryptovaranoides microlanius *in (above) anterior and (below) posterior views* **showing similar features except the condyle of the capitellum**'. (boldface added). Then, in *Whiteside et al., 2024* (subsection 2.3), the authors stated that this fragment does not have a radial condyle/capitellum of the humerus, which they defend as follows: '*This is reinforced by the larger humerus (*Figure 1b*) which is missing the condyle,* **as is typical in the preservation of fissure lepidosaurs** *(e.g.* †Clevosaurus; *Smith et al., 2020, fig. 29b)* **but the cavity in which it sat clearly indicates a substantial condyle in life***'. (boldface added). What *Whiteside et al., 2024* highlighted is that there is a radial condyle (i.e. capitellum) preserved in some specimens, but not in others, deferring such differences among the materials referred to †*Cryptovaranoides* to poor preservation without justification beyond '*as is typical ….of fissure lepidosaurs…*'. Taphonomy and poor preservation cannot be used to justify the inference that an anatomical feature was present when it is not preserved and when there is no evidence of postmortem damage. In such a situation, when a feature's absence is potentially ascribable to preservation, its presence should be considered ambiguous—or that some of these materials may simply not belong to the same taxon, which is the point raised previously by *Brownstein et al.,*

*2023*. Finally, even in the case of the isolated humerus with a preserved radial condyle illustrated by *Whiteside et al., 2024*, this is fairly small compared to that of squamates, either crown members or the earliest known pan-squamates, such as †*Megachirella wachtleri* (Figure 1 in *Simões et al., 2018*).

## Evaluation of character state determinations provided by *Whiteside et al., 2024*

In this section, we revisit each alternative interpretation of the anatomy of †*Cryptovaranoides microlanius* provided by *Whiteside et al., 2024*, who discussed 26 characters that they suggest have bearing on the placement of this taxon within Lepidosauria, Pan-Squamata, crown Squamata, and successive clades within the crown. We note first that these characters do not correspond to optimized character states in the phylogenies that (*Whiteside et al., 2022*; *Whiteside et al., 2024*) infer, but instead to an assemblage of character state optimizations presented in various papers in the literature (e.g. *Gauthier et al., 2012*; *Estes et al., 1988*; *de Queiroz and Gauthier, 2020a*). This is not an issue per se for a referral of †*C. microlanius* to Squamata, but it does mean that these characters are of unclear relevance to the actual support for the position(s) of †*C. microlanius* among reptiles that (*Whiteside et al., 2022*; *Whiteside et al., 2024*) recovered in the phylogenies that they present. With this noted, we provide a point-by-point discussion of character interpretations in *Brownstein et al., 2023* that *Whiteside et al., 2024* challenged.

### Preservation of the septomaxilla

*Whiteside et al., 2022*; *Whiteside et al., 2024* claimed that a small, disarticulated piece of bone preserved anterolateral to the vomer in block containing the holotype of †*Cryptovaranoides microlanius* is the septomaxilla. They further suggested that a portion of the medial surface on a maxilla referred to this taxon but clearly from a much larger reptile provides additional support for the presence of a septomaxilla in †*C. microlanius*. The CT scans published by *Whiteside et al., 2022*; *Whiteside et al., 2024* show that the bone in the holotype NHMUK PV R36822 is isolated and with no clear morphological affinities to the septomaxillae in squamates (see CT scans in *Gauthier et al., 2012*). *Whiteside et al., 2024* also suggested that the identity of this bone as the septomaxilla is supported by its placement between the maxilla and premaxilla of the holotype, but given the level of disarticulation of the holotype and the morphology of the bone fragment, a similar argument could be made that it is a portion of the anterior end of one of the vomers. In any case, we feel it is premature to code †*C. microlanius* for any character related to the morphology of the septomaxilla based on this disarticulated and damaged bone fragment.

### Expanded radial condyle of the humerus

*Whiteside et al., 2024* provided additional justification of the presence of an expanded radial condyle of the humerus by stating that the projection of the radial condyle above the adjacent region of the distal anterior extremity of †*Cryptovaranoides microlanius* supported their choice to score the expanded radial condyle as present. The projection of the radial condyle above the adjacent region of the distal anterior extremity is not the condition specified in either of the morphological character sets that (*Whiteside et al., 2024*) cited (*Brownstein et al., 2023*; *Tałanda et al., 2022*). The condition specified in those studies is the presence of a distinct condyle that is expanded. The feature described in *Whiteside et al., 2024* does not correspond to the character scored in the phylogenetic datasets (see also additional considerations for this character in the previous section).

### Absence of the posterior process of the jugal

*Whiteside et al., 2024* suggests that the absence of the posterior process of the jugal in †*Cryptovaranoides microlanius* supports its affinities to Squamata. The absence of the jugal posterior process, which we have argued is not clear in the holotype of this species and may be worn off (*Brownstein et al., 2023*), is a notoriously variable character among lepidosaurs that has caused substantial confusion about the placement of fossils. The new figure provided by *Whiteside et al., 2024*; *Figure 1f* of the jugal is pixelated and unclear. The presence of a jugal posterior process, including its development into a complete jugal bar, is documented for numerous pan-squamate, pan-lepidosaur, and extinct crown lepidosaur species (*Simões et al., 2018*; *Gauthier et al., 2012*; *Mo et al., 2010*;

*Tałanda et al., 2022*; *Ford et al., 2021*; *Griffiths et al., 2021*). Notably, and as stated by *Whiteside et al., 2024*, complete loss of the temporal bar potentially diagnoses all lepidosaurs (with presence in rhynchocephalians being a reversal), so absence of the jugal process in †*Cryptovaranoides microlanius* (if confirmed) cannot be used to place it within Squamata. The lower temporal bar is, however, also incomplete, and the jugal posterior process is variably developed in numerous clades within Archosauromorpha, including the †Azendohsauridae (*Sengupta and Bandyopadhyay, 2021*; *Sengupta et al., 2017*; *Flynn et al., 2010*; *Sen, 2003*), †*Prolacerta broomi* (*Modesto and Sues, 2004*; *Sobral, 2023*), †*Teyujagua paradoxa* (*Pinheiro et al., 2016*), and '†Protorosauria' (*Miedema et al., 2020*; *Spiekman et al., 2021*; *Spiekman et al., 2020a*; *Nosotti, 2007*; *Spiekman et al., 2020b*; *Spiekman et al., 2023*). The development and enclosure of the temporal fenestra in reptiles has been linked to the expression of two genes, *Runx2* and *Msx2*, in in vivo studies (*Tokita et al., 2013*). We suspect that the variable extent of this feature in many early-diverging lepidosaur and archosauromorph lineages might be an example of the 'zone of variability' (*Bever et al., 2011*), in which the canalization of development and other constraints (e.g. functionality; see *Schaerlaeken et al., 2008*) had not yet completely acted to 'fix' the morphology of the posterior process. This hypothesis will, of course, require further experimental study of living model systems.

## Anterior emargination of the maxillary nasal process

*Whiteside et al., 2024* flagged our interpretation of the anterior maxillary nasal process as emarginated, which we found united †*Cryptovaranoides microlanius* with archosauromorphs. Although we still interpret this state as such based on the computed tomography scan data, we again note that none of our analyses (*Brownstein et al., 2023*) unambiguously place †*Cryptovaranoides microlanius* within Archosauromorpha and that, while optimized as such, this single character necessarily has limited utility for placing †*C. microlanius* among reptiles.

## Subdivision of the metotic fissure

*Whiteside et al., 2024* claimed that the division of the metotic fissure into the vagus foramen and recessus scalae tympani by the crista tuberalis, a key squamate feature, can be scored for †*Cryptovaranoides microlanius*, yet paradoxically suggested the presence of this condition is only inferable based on other observations of the anatomy of the holotype and referred specimens. Actually, *Whiteside et al., 2024* argued that because another character relating to a different part of the structure of the metotic fissure can be inferred based on the presence of the crista tuberalis, the division of the metotic fissure may also be inferred by the presence of the crista tuberalis. This logic would imply that no reptile taxon should exist that possesses solely either a crista tuberalis or a subdivided metotic fissure, even when this is an observed state combination in squamates scored for the matrices that both our teams have used to analyze the phylogenetic position of †*Cryptovaranoides microlanius* (*Simões et al., 2018*; *Tałanda et al., 2022*). This inference resultantly lacks any justification. In an attempt to verify that †*C. microlanius* possessed a vagus foramen, *Whiteside et al., 2024* stated that they searched for isolated otoccipital fragments from the same locality and found an otoccipital fragment (NHMUK PV R36822) with a vagus foramen that they refer to †*C. microlanius*. It is unclear to us how *Whiteside et al., 2024Whiteside et al., 2024* accounted for confirmation bias when conducting this collections search, or on what basis they refer this isolated bone to †*C. microlanius*. In any case, the presence of the lateral opening of the recessus scalae tympani is not observable in the fragment, and thus the division of the metotic fissure is not demonstrated in the referred fragment.

## Fusion of exoccipitals and opisthotics

As *Whiteside et al., 2024* noted, we concur with their identification of an otoccipital in †*Cryptovaranoides microlanius* that is formed by the fusion of the exoccipitals with the opisthotics. Our concern is with the use of this feature to assign †*C. microlanius* to Squamata. *Whiteside et al., 2024* cited *de Queiroz and Gauthier, 2020a*, who list the presence of an otoccipital as a distinguishing feature of squamates. However, *Whiteside et al., 2024* failed to provide any phylogenetic evidence supporting the optimization of this feature as a squamate synapomorphy in phylogenies including †*C. microlanius*, nor did they recognize that an otoccipital is present in numerous non-squamate reptiles, including numerous archosauromorphs (*Marsh et al., 2022*; *Ezcurra, 2016*; *Pinheiro et al., 2020*). For these reasons, the presence of an otoccipital cannot be used to assign †*C. microlanius* to Squamata or even

Lepidosauria instead of Archosauromorpha or other clades of reptiles known from the Permo-Triassic, except probably the turtle total clade (†*Eunotosaurus africanus* possesses unfused exoccipitals and opisthotics *Bever et al., 2015*). We reiterate that *Whiteside et al., 2024* did not show that the presence of an otoccipital optimizes as an ambiguous or unambiguous synapomorphy of Lepidosauria, Pan-Squamata, or Squamata in any of their phylogenies. For characters that show evidence of homoplastic evolution like the presence of an otoccipital (and indeed, the presence of a jugal posterior process; see above), phylogenetic character optimization is essential.

## Enclosed vidian canal exiting anteriorly at base of each basipterygoid process

In our restudy of the holotype of †*Cryptovaranoides microlanius*, *Brownstein et al., 2023* were unable to verify the presence of an enclosed vidian canal exiting the sphenoid (=parabasisphenoid) via the base of the corresponding basipterygoid process. *Whiteside et al., 2024* took issue with *Brownstein et al., 2023* interpretation of this region of the braincase and suggested that a larger, abraded, and isolated sphenoid (NHMUK PV R 37,603 a) that they unjustifiably refer to †*C. microlanius* supports their interpretation of the anatomy of this region of the braincase, yet they also say that this fragment is of only limited informativeness and then they appear to agree with us that the best course of action is to score this character as missing data when including †*C. microlanius* in phylogenetic analyses.

## Development of the choanal fossa of the palatine

Squamata includes multiple instances where the complexity of the bony palate increases dramatically, an innovation that is related to chemosensory evolution and the integration of different bones that form this region of the skull (*Gauthier et al., 2012*; *Gauthier et al., 2012*; *Estes et al., 1988*; *Brownstein et al., 2022*; *Rieppel et al., 2009*; *Lee, 1997*; *Abramyan and Richman, 2015*; *Strong et al., 2022*; *Watanabe et al., 2019*). One feature recognized as phylogenetically informative is the development of the choanal fossa on the ventral surface of the palatine (also known as the palatine sulcus; *Brownstein et al., 2023*; *Gauthier et al., 2012*; *Brownstein et al., 2022*; *Meyer et al., 2023*). *Whiteside et al., 2024* claim that the development of the palatine choanal fossa in †*Cryptovaranoides microlanius* is comparable to the development of this feature in living squamates, and cite the living iguanian genus *Ctenosaura* as an example of a squamate with similar anteroposterior development of this feature as †*C. microlanius*. Importantly, most iguanians appear to show an apparently plesiomorphic condition where the choanal fossa is anteriorly restricted on the palatine *Gauthier et al., 2012*; this feature has actually contributed to debates about whether iguanians form the living sister to all other crown squamates (as found in some morphological character-based phylogenies; *Gauthier et al., 2012*; *Estes et al., 1988*) or are deeply nested within the crown clade and secondarily convergent with rhynchocephalians. (as implied by phylogenies based on DNA sequence data *Title et al., 2024*; *Burbrink et al., 2020*; *Townsend et al., 2004*; *Pyron et al., 2013*; *Streicher and Wiens, 2017*; *Zheng and Wiens, 2016*; *Singhal et al., 2021* alone or with morphological data *Simões et al., 2018*). Some crown squamates almost completely lack the fossa (Figure 2; Figure 3), further complicating the argument that the choanal fossa in †*C. microlanius* represents the plesiomorphic squamate condition; a proper phylogenetic analysis is needed.

Secondly, *Whiteside et al., 2024* took issue with *Brownstein et al., 2023* contention that the palatine fossa is present, albeit variably, across a wide swath of reptilian diversity, including many early-diverging archosauromorph clades. They explicated this concern by distinguishing between the narrow channel found in taxa like †*Tanystropheus* (figure 1l in *Whiteside et al., 2024*) and the wider fossa found in †*C. microlanius*. We agree that the fossa in †*C. microlanius* is more developed than in some early-diverging lepidosaurs, such as †*Marmoretta oxoniensis* (*Griffiths et al., 2021*). However, it is clear that the feature in †*Cryptovaranoides* falls within the variation in the choanal fossa length and depth (*Figure 2*; *Figure 3*) that represents the ancestral condition in lepidosaurs *Gauthier et al., 2012* and is exemplified across archosauromorph (e.g. *Marsh et al., 2022*) and indeed diapsid (*Dudgeon et al., 2020a*; *Brownstein, 2022*; *Jenkins et al., 2024*) diversity. Finally, (*Whiteside et al., 2024*) cited the discovery of a large, isolated palatine that they stated confirmed the presence of a 'squamate-type' choanal fossa in †*C. microlanius*, except they provided no justification for why this isolated bone should be referred to this species. In any case, the morphology of that bone is also unlike those of squamates (*Figure 2*).

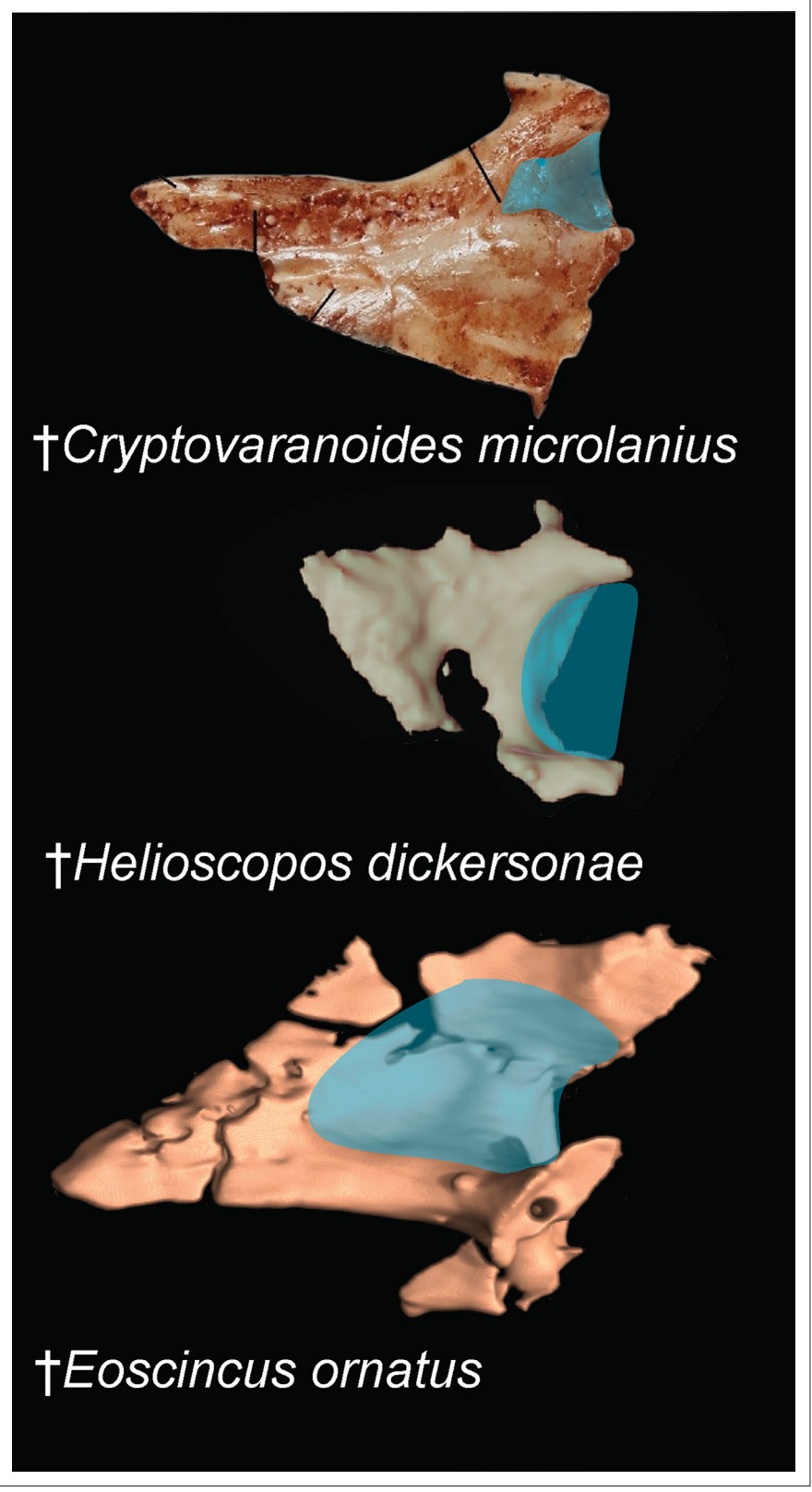

**Figure 2.** Comparison of palatine morphologies. Blue shading indicates choanal fossa. Top image of †*Cryptovaranoides* referred left palatine is from *Whiteside et al. (2024)* Figure 1(k). Middle is the left palatine of †*Helioscopos dickersonae* (Squamata: Pan-Gekkota) from the Late Jurassic Morrison Formation (*Meyer et al., 2023*). Bottom is the right palatine of †*Eoscincus ornatus* (Squamata: Pan-Scincoidea) from the Late Jurassic Morrison Formation (*Brownstein et al., 2022*).

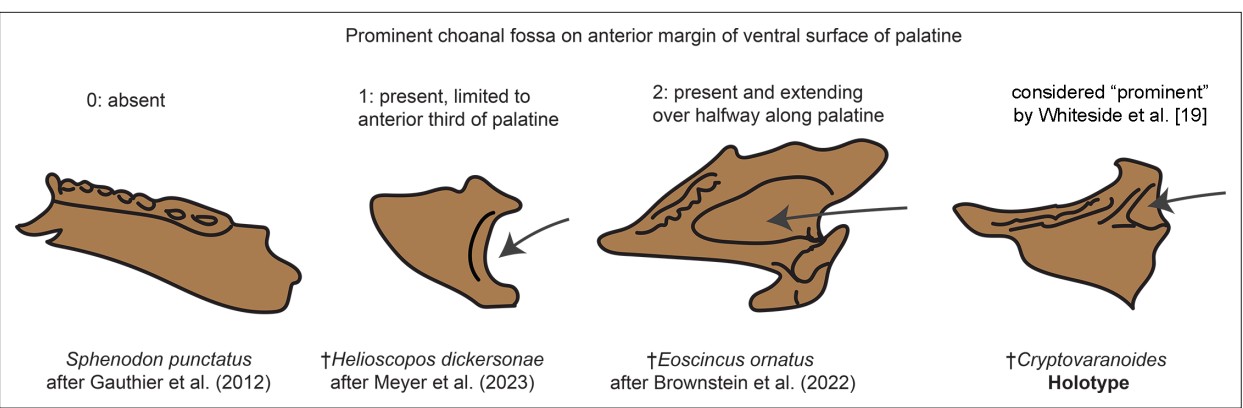

Figure 3. Choanal fossa character scoring.

## Vomer ventral ridges and dentition

*Whiteside et al., 2024* reiterated their earlier characterization (*Whiteside et al., 2022*) of the vomer of †*Cryptovaranoides microlanius* as ridged, and suggested that the morphology of the vomerine ridges and vomerine teeth in †*C. microlanius* is comparable to the condition in the anguid anguimorph *Pseudopus apodus*. We dispute this on the basis that the vomer of †*C. microlanius* as figured by *Whiteside et al., 2024* showed no clear ridges equivalent to those in *Pseudopus apodus* (*Klembara et al., 2017*) or other squamates with ridged vomers, including extinct forms such as †*Eoscincus ornatus* (*Brownstein et al., 2022*; *Figure 4*). Instead, the new photograph of the holotype specimen of †*C. microlanius* provided in *Whiteside et al., 2024* showed that the vomer is indeed toothed, but no ridges are figured or visible. We once again re-examined the CT scan data and failed to find any structure resembling the ridges present in some anguimorphs, although we acknowledge that the resolution of the scan (43 μm per voxel) prevents us from categorically stating that lower ridges cannot be present. In any case, no ridges equivalent in shape or size to those found in anguimorph squamates are present in the holotype of †*C. microlanius.*

The presence of teeth on the vomer is itself rare among squamates; only in the pan-scincoid †*Eoscincus ornatus Brownstein et al., 2022* and some (*Klembara et al., 2017*) anguid and varanoid (*Yi and Norell, 2013*) species are vomerine teeth documented. *Whiteside et al., 2022*; *Whiteside et al., 2024* appeared to suggest that the presence of vomerine ridges and a row of vomerine teeth therefore allies †*C. microlanius* with anguimorphs, even though none of the phylogenies that they infer (and indeed, no phylogeny to our knowledge) optimizes the presence of vomerine teeth as a synapomorphy of crown Anguimorpha. Furthermore, the structure of the vomer in †*C. microlanius* is fundamentally unlike those of anguimorph squamates, which are elongate and possess pronounced ridges that are placed medially on the ventral surface of the main body of each vomer and each house only one row of teeth along their posterior third (*Klembara et al., 2017*; *Smith and Habersetzer, 2021*). In contrast, the condition that *Whiteside et al., 2024* figured for †*C. microlanius* shows multiple, parallel rows of vomerine teeth on either side of the vomer that run across the entire length of the bone. This morphology is even unlike that observed in the only squamate known where multiple vomerine tooth rows are present, †*Eoscincus ornatus* (*Brownstein et al., 2022*; *Figure 4*). Multiple rows of vomerine teeth are common in reptiles outside of Squamata *Matsumoto and Evans, 2017*; the presence of only one row is restricted to a handful of clades, including millerettids (*Jenkins et al., 2025a*; *Jenkins et al., 2025b*), †*Tanystropheus* (*Spiekman et al., 2020b*), and some (*Dudgeon et al., 2020b*), but not all (*Brownstein, 2022*; *Matsumoto and Evans, 2016*) choristoderes. In †*Eoscincus ornatus*, all vomerine teeth are restricted to a raised surface at the posterior end of the ventral surface of the main body of the vomer (*Brownstein et al., 2022*). Thus, *Whiteside et al., 2024* provided no evidence that vomerine ridges are present in †*C. microlanius* nor that the presence of ridges and teeth is a synapomorphy of Anguimorpha.

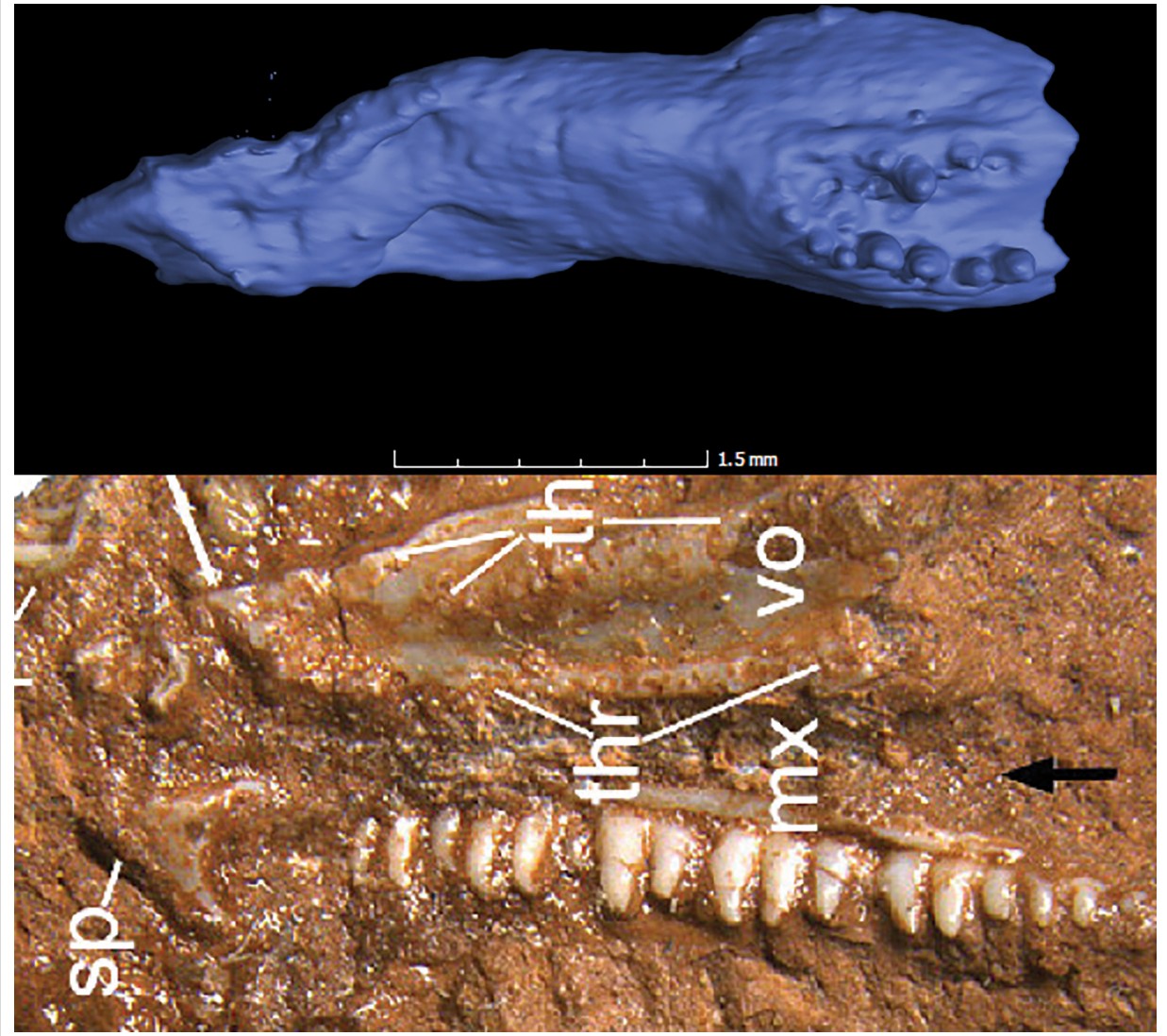

**Figure 4.** Comparison of vomer morphologies. Top image is †*Eoscincus ornatus* from **Brownstein et al., 2022**. Note that the rows of vomerine teeth are posteriorly placed, and the vomerine ridges are large and laterally placed. Bottom image of †*Cryptovaranoides* holotype is from **Whiteside et al. (2024)** Figure 1(c).

### Lacrimal arches dorsally over lacrimal duct and floors lacrimal duct with medial process posteriorly

*Brownstein et al., 2023* argued that this feature was unobservable in the holotype of †*Cryptovaranoides microlanius* (**Brownstein et al., 2023**). *Whiteside et al., 2024* disputed this by suggesting that their CT scan data does not reliably show this feature and instead provided a photograph of the holotype skull that they stated shows this morphology (figure 2a-c in *Whiteside et al., 2024*). This figure shows no curvature to the lacrimal, which appears as a flat element, whereas an arch dorsally and a floor ventrally would indicate the presence of a foramen. We are also confused by *Whiteside et al., 2024* statement that they did not code this feature (dorsal arcuation of the lacrimal over the lacrimal duct, which is posteriorly floored by the medial process of the lacrimal) for †*C. microlanius* but still used it to refer this species to the Anguimorpha based on its optimization as a plesiomorphy of that clade in phylogenetic analyses that place †*C. microlanius* within Anguimorpha. This appears to suggest that, despite never conducting an analysis where this feature is coded as 'present' for †*C. microlanius*, *Whiteside et al., 2024* used the presence of this feature to ally †*C. microlanius* with Anguimorpha not as a recovered synapomorphy but from merely deciding it was so. This action would

be an arbitrary unification of a species with a clade based on a selected character state and would deny the equal possibility that this character state is absent due to secondary reversal in †*C. microlanius*. In sum, the use of lacrimal morphology to ally †*C. microlanius* with anguimorphs is apparently not based on direct character state optimization, that is a test of congruence.

## Distinct quadratojugal absent

The quadratojugal cannot be located in the holotype of †*Cryptovaranoides microlanius* (**Whiteside et al., 2022**; **Whiteside et al., 2024**). **Brownstein et al., 2023** argued that this might be due to post-mortem disarticulation and damage to the skull, and also noted that a complete ontogenetic series would *ideally* be required to test whether the quadratojugal changes shape throughout ontogeny (**Brownstein et al., 2023**). **Whiteside et al., 2024** focused on our comment about the ideal situation of having an ontogenetic series for †*C. microlanius* to assess the development of the quadratojugal, which we restate would be helpful to understand how this bone transforms through ontogeny given the complex restructuring to this region of the skull that occurs throughout the evolution of lepidosaurs (**Ford et al., 2021**). However, **Whiteside et al., 2024** stated, 'we argue based on juvenile and adult specimens and the absence of a quadratojugal facet on the quadrate'.

First, as highlighted by **Brownstein et al., 2023**, the region of the quadrate that would articulate with the quadratojugal is not preserved in the holotype of †*C. microlanius*, so it is impossible to tell whether a facet for the quadratojugal is present. Second, **Whiteside et al., 2024** fail to provide any justification for the referral of the isolated partial quadrate NHMUK PVR 37606 to †*C. microlanius*. Third, **Whiteside et al., 2024** fail to identify the quadratojugal facet on this isolated quadrate when they figure this bone; indeed, based on the figure, the process that would have housed the quadratojugal facet is also missing from this quadrate (NHMUK PVR 37606). It is impossible to tell whether †*C. microlanius* lacked a quadratojugal based on the current data. Instead, all that can be said is that a quadratojugal is not preserved in the holotype of †*C. microlanius*, and the region housing the articular facet for the quadratojugal is not preserved in either the holotype or referred quadrate.

## Pterygoid/quadrate overlap

*Whiteside et al., 2024* restate their original interpretation of *Whiteside et al., 2022* that the pterygoid and quadrate have a short overlap in †*Cryptovaranoides microlanius*. *Whiteside et al., 2024* oddly restrict comparisons of the morphology of the quadrate in †*Cryptovaranoides microlanius*, which they agree is damaged, to the morphology present in rhynchocephalians and suggest based on this comparison that their interpretation is correct. We restate that this is not possible to verify without more complete, articulated, or semi-articulated palates assignable to †*Cryptovaranoides microlanius* based on apomorphic character states and combinations.

## Fusion of the premaxillae and single median tooth

*Whiteside et al., 2024* suggest that *Brownstein et al., 2023* incorrectly characterized the nature of fusion of the premaxillae into a single median element in †*Cryptovaranoides microlanius*. However, we reiterate that no justification has been given in their studies *Whiteside et al., 2022*; *Whiteside et al., 2024* for referring these isolated premaxillae to †*C. microlanius*. *Whiteside et al., 2024* focus on differentiating these isolated large premaxillae from the premaxillae of two other fissure fill lepidosaurs, †*Gephyrosaurus bridensis* and †*Diphydontosaurus avonis*, apparently without considering the possibility that additional taxa are present in the assemblage. Scoring fused premaxillae as present for †*C. microlanius* despite the presence of unfused, paired premaxillae in the holotype defines (1) ontogenetic character state transformations solely based on the relative size of the holotype and larger isolated bones referred without justification and (2) which character state among those present in ontogeny is the phylogenetically informative one. Defining both of these requires a robust ontogenetic series, which is not available for †*C. microlanius* at present. Observations of the development of living squamates also suggest that these larger fused premaxillae should not be referred to †*C. microlanius*. In squamates with a median premaxilla, the premaxilla is invariably a single element upon first appearance very early in embryonic development (**Khannoon and Evans, 2020**; **Hernández-Jaimes et al., 2012**; **Skawiński et al., 2021**; **Skawiński et al., 2023**; **Ollonen et al., 2018**). In contrast, the holotype of †*C. microlanius*, which is very clearly a juvenile (i.e. post-embryonic) specimen, possesses

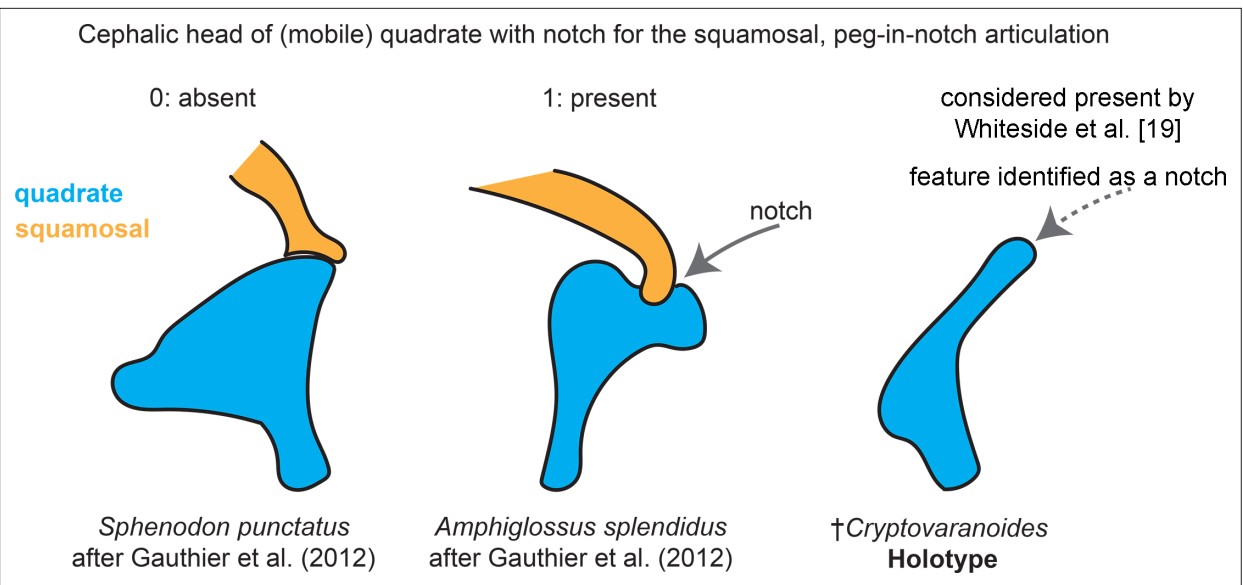

**Figure 5.** Quadrate-squamosal articulation character scoring.

paired premaxillae. The ontogenetic series that *Whiteside et al., 2024* suggest for †*C. microlanius* therefore differs from any known squamate with a single median premaxilla.

## Peg-in-notch articulation of quadrate with rod-shaped squamosal

*Whiteside et al., 2024* suggest that both their study and *Brownstein et al., 2023* are in agreement about the presence of a peg-in-notch articulation between the quadrate and squamosal. However, *Brownstein et al., 2023* suggested (and we reiterate herein) that the presence of this type of articulation is unclear based on the available data for the holotype, as these bones are both damaged in the relevant sections and disarticulated (*Figure 5*).

## Frontal underlaps parietal laterally on frontoparietal suture

*Whiteside et al., 2024* confirm that the presence of this feature in †*Cryptovaranoides microlanius* is entirely based on the morphology of a referred isolated frontal that they state matches the corresponding articular portion of the prefrontal in the holotype. However, we do not understand how this element could match the corresponding articular surface unless it was from the same individual or an animal of the same size. Unless *Whiteside et al., 2024* were able to articulate these bones, it is unclear how this inference can be made. Further, we note that the posterior process of the prefrontal is broken off in the holotype of †*Cryptovaranoides microlanius*. In squamates and other lepidosaurs, this process abuts the lateral surface of the frontal along its anteroposterior axis. Two of us (C.D.B. and D.L.M.) worked to rearticulate the holotype of †*Eoscincus ornatus*, which involved rearticulating the prefrontal and frontal (*Brownstein et al., 2022*). Rearticulating these bones is simply impossible without the complete posterior process of the prefrontal, as the orientation of this process must match the curvature of the lateral margin of the frontal. Because the isolated frontal is not figured in either paper by *Whiteside et al., 2022*; *Whiteside et al., 2024*, it is impossible for us to verify whether this bone actually matches the corresponding articulation surface on the prefrontal in the holotype. In any case, we regard it as a best practice to exclude this isolated frontal from discussions of the affinities of †*C. microlanius*.

## Medial process of the articular and prearticular

*Whiteside et al., 2024* revise their initial (*Whiteside et al., 2022*) characterization of the medial process and refer to it as 'rudimentary,' scoring it as unobservable. We determined that this process is absent (*Brownstein et al., 2023*) and offer no further comment besides that it appears the distinction

between our interpretation of this feature and that of *Whiteside et al., 2024* is, as they explicated, largely arbitrary.

## Bicapitate cervical ribs and cervical ribs with an anteriorly oriented process

*Whiteside et al., 2024* took issue with *Brownstein et al., 2023* characterization of the morphology of the cervical ribs in †*Cryptovaranoides microlanius* and compared the bicapitate morphology of the cervical ribs in †*C. microlanius* to two anguimorph squamates, *Pseudopus apodus* and *Varanus* spp. However, the cervical ribs of *P. apodus* are not bicapitate, as indeed shown by the figure from *Cerňanský et al., 2019* that (*Whiteside et al., 2024*) cited. Similarly, the ribs of *Varanus* are unicapitate, not bicapitate, as shown by *Cieri, 2018*. We assume that *Whiteside et al., 2024* referred to the morphology of the cervical rib head in *P. apodus* because of the presence of a posterior process on the rib head (*Cerňanský et al., 2019*). This is not the bicapitate condition and is not homologous with the morphology of the cervical rib heads in either †*C. microlanius* or archosauromorphs, where an anterior process is offset from the process formed by the capitulum and tuberculum (*Ezcurra, 2016*; *Nesbitt, 2011*). We reiterate that the condition in †*C. microlanius* is identical to that in archosauromorphs (Figures 2C and 3 in *Brownstein et al., 2023*); the comparisons made by *Whiteside et al., 2024* are across non-homologous structures and result from a misinterpretation of cervical rib anatomy. This confusion in part stems from the lack of a fixed meaning for uni- and bicapitate rib heads; in any case, †*C. microlanius* possesses a condition identical to archosauromorphs as we have shown.

## Cervical and dorsal vertebral intercentra

*Whiteside et al., 2024* suggest that *Brownstein et al., 2023* inferred the presence of cervical and dorsal intercentra in †*Cryptovaranoides microlanius*; however, we did not propose this in that study. In fact, *Whiteside et al., 2022* state in their original paper that there 'are gaps between the vertebrae indicating that intercentra were present (but displaced in the specimen) on CV3 and posteriorly. Some images of bones on the scans are identified as intercentra' (p. 11, *Whiteside et al., 2022*). The statement in *Whiteside et al., 2024* consequently represents an incorrect attribution and an incorrect characterization of our reexamination, as we determined that no cervical and dorsal intercentra were preserved or present in the holotype. *Whiteside et al., 2024* justify their position that cervical intercentra are present in †*C. microlanius* based on the morphology of another isolated bone that they refer to this taxon without justification. In any case, the absence of dorsal intercentra is not a distinguishing feature of squamates because several lineages of squamates, including lacertids, xantusiids, gekkotans, and the stem-squamate †*Bellairsia gracilis,* all possess cervical and dorsal intercentra (*Tałanda et al., 2022*; *Barbadillo and Martínez-Solano, 2002*; *Hoffstetter, 1969*).

## Anterior dorsal vertebrae, diapophysis fuses to parapophysis

*Whiteside et al., 2024* concur with *Brownstein et al., 2023* that the diapophyses and parapophyses are unfused in the anterior dorsals of the holotype of †*Cryptovaranoides microlanius*, and restate that fusion of these structures is based on the condition they observed in isolated vertebrae (e.g. NHMUK PV R37277) that they refer to †*C. microlanius* based on general morphological similarity and without reference to diagnostic characters of †*C. microlanius*. This feature should not be scored as present for †*C. microlanius*.

## Zygosphene–zygantrum in dorsal vertebrae

*Whiteside et al., 2024* again claim that the zygosphene-zygantrum articulation is present in the dorsal series of †*Cryptovaranoides microlanius* based on the presence of 'rudimentary zygosphenes and zygantra' in isolated vertebrae (NHMUK PV R37277) that they again refer to this species without justification. The structures that *Whiteside et al., 2024* label as the zygosphenes and zygantra (figure 3 in *Whiteside et al., 2024*) are clearly not, however, zygosphenes and zygantra, as a zygosphene is by definition a centrally located wedge-like process that fits into the zygantrum, which is a fossa on the following vertebra. The structures labeled zygosphenes and zygantra by *Whiteside et al., 2024* are just the dorsal surfaces of the prezygapophyses and the medial margins of the postzygapophyses.

## Anterior and posterior coracoid foramina/fenestra

*Whiteside et al., 2024* used isolated coracoids (e.g. NHMUK PV R37960) referred to †*Cryptovaranoides microlanius* without justification to support their claim that the upper 'fenestra' (foramen) of the coracoid in the holotype specimen is indeed a foramen. Confusingly, *Whiteside et al., 2024* labeled this feature as the 'primary coracoid fenestra' in their Figure 3i–j, even though it is clearly the coracoid foramen. *Whiteside et al., 2024* appear to have confused the coracoid foramen, which is completely bounded by bone and placed inside the coracoid, with the coracoid fenestra, which is bounded *in part* by the coracoid (the coracoid margin is curved to form part of the bounding region in species with the coracoid fenestra) but also by the interclavicle (see figures in *Brownstein et al., 2023*; *Gauthier et al., 2012*). The coracoid fenestra is not contained within the coracoid. This feature is also not shown to be optimized as a pan-squamate synapomorphy in any phylogeny including †*Cryptovaranoides microlanius* (*Whiteside et al., 2022*; *Whiteside et al., 2024*), so its bearing on the identification of †*C. microlanius* as a pan-squamate is unclear to us. Finally, we note here that *Whiteside et al., 2024* appear to have labeled a small piece of matrix attached to a coracoid that they refer to †*C. microlanius* as the supracoroacoid [*sic*] foramen in their Figure 3, although this labeling is inferred because only 'suc, supracoroacoid [*sic*]' is present in their caption.

## Atlas pleurocentrum fused to axis pleurocentrum

*Whiteside et al., 2024* reiterated their claim that the atlas and axis pleurocentra are present in †*Cryptovaranoides microlanius* based on their identification of an isolated, globose bone fragment (NHMUK PV 38897) as the atlas intercentrum, but provide no additional justification for this identification and refer the reader to their original paper for a figure of this bone (*Whiteside et al., 2022*), which is only visible on CT scans due to its entombment within the matrix that includes the holotype. To this end, *Whiteside et al., 2024* revised their initial interpretation of what they identified as the preserved atlas-axis region, but again without any justification or figures detailing how they reidentified a bone fragment that they had initially believed was a cervical intercentrum as the atlas centrum and intercentrum 2. Thus, *Whiteside et al., 2024* did not provide any additional description of how they reidentified these bones or a figure illustrating their revised interpretation, and so we cannot comment on the strength of their revised interpretation. Again, we note that we could not identify the morphology of the atlas and axis with any confidence in the holotype of †*C. microlanius*, and so we again believe any relevant characters should be scored as missing data for this taxon.

## Midventral crest of presacral vertebrae

*Whiteside et al., 2024* appeared to agree with *Brownstein et al., 2023* assertion that a midventral crest is not present on the presacral vertebrae (*Brownstein et al., 2023*). We never challenged their scoring of keels on the caudal centra as missing data. Rather, we stated that keels are clearly present on the cervical vertebrae (*Brownstein et al., 2023*).

## Angular does not extend posteriorly to reach the articular condyle

*Whiteside et al., 2024* stated that the posterior portion of the angular is present medially on the right mandible in the holotype and provided an interpretation of the anatomy of this region. They cited figure 3a in *Whiteside et al., 2024*, but this shows the mandible in lateral view and only a small portion of the angular is visible. As such, it is not clear to us what they interpreted to be the posterior portion of the angular. In any case, we could not identify the feature that *Whiteside et al., 2024* considered to be the posterior angular extent anywhere on the CT scans or on the accessible portions of the real specimen (e.g. figure 2b in *Whiteside et al., 2024*). *Whiteside et al., 2024* described the posterior extent of the angular as a 'contoured feature', but we are entirely unclear about what this actually describes.

## Ulnar patella

The only disagreement between *Brownstein et al., 2023* interpretation and that made by *Whiteside et al., 2022*; *Whiteside et al., 2024* concerns whether the ulnar patella is absent due to the ontogenetic state or preservation status of the holotype (*Whiteside et al., 2022*; *Whiteside et al., 2024*) or an ontogenetically invariable feature of the anatomy of †*Cryptovaranoides microlanius* (*Brownstein*

*et al., 2023*). We suggested the latter based on our observation that the forelimb of the holotype is articulated and mostly complete.

## Overarching empirical problems in *Whiteside et al., 2024*

In this section, we focus on broader empirical issues in *Whiteside et al., 2024* that directly impact conclusions regarding the taxonomy and phylogenetic placement of *Cryptovaranoides*.

### Unassignable specimens and 'hypodigm inflation'

*Whiteside et al., 2022* and *Whiteside et al., 2024* described and defended their addition of isolated elements to fill in gaps in the holotype concept of †*Cryptovaranoides*; in our view, this creates an unjustifiably inflated hypodigm. *Whiteside et al., 2024* (p. 15) try to ameliorate this problem in their Section (5.5) where they state that: '*We emphasize that we always match isolated bones with their equivalents on the holotype*' and then state in contradiction that: '*We consider it **remiss** not to include isolated bones as they provide details which no scan can. Furthermore, as the holotype is a juvenile, they give additional information from larger, and presumably older individuals, **on characters of the holotype otherwise unavailable or uncertain**'*. (boldface added). In summary, *Whiteside et al., 2024* claimed to only match isolated bones with equivalent ones in the holotype but also acknowledged use of larger elements that are not comparable with the ones in the holotype, but the morphological justification for those referrals (e.g. shared autapomorphies) was not always explicit. This lack of consistency is a serious empirical issue.

### Apomorphic characters not empirically obtained

*Whiteside et al., 2024* discussed nearly 30 anatomical characters (Sections 2–6), most of which they used to support their core hypothesis that †*Cryptovaranoides* is a squamate. Yet, none of these 30 characters or conditions are actually found by *Whiteside et al., 2024* to be synapomorphies of squamates in their various phylogenetic analyses. Section 6, (*Whiteside et al., 2024*) (p. 16) indicated they modified the datasets of *Brownstein et al., 2023* and *Tałanda et al., 2022*, but preferred the results of *Budd and Mann, 2024* and performed two different analyses based on *Tałanda et al., 2022*: one using only morphological data, and one using morphological data with several constraints from a molecular backbone. The former was tested using maximum parsimony and the latter using Bayesian inference. What is not clear, from either Section 6 or 7, is which phylogenetic analysis was used by *Whiteside et al., 2024* to '*review the apomorphy distribution*'. But this uncertainty is inconsequential as the apomorphy distributions discussed in Section 7 of *Whiteside et al., 2024* were not derived from their phylogenetic analyses. Instead, *Whiteside et al., 2024* (p. 19) exclusively extracted characters from the literature to support their identification of †*Cryptovaranoides* as a squamate, rather than basing this inference primarily on their own phylogenetic analyses. These literature-sourced characters include '*two diagnostic characters of Squamata (=crown-clade Squamata)*' and '*further eight squamate synapomorphies*' listed by *Whiteside et al., 2024* (Section 7, p. 19). As stated by the authors, '*we give the citation for the squamate synapomorphy at the end of each character*', indicating that these diagnostic characters or synapomorphies were picked from the literature and not derived from ancestral–state reconstructions based on their phylogenetic results.

In order to check the characters listed by *Whiteside et al., 2024* (p. 19) as '*two diagnostic characters*' and '*eight synapomorphies*' in support of a squamate identity for †*Cryptovaranoides*, we conducted a parsimony analysis of the revised version of the dataset (*Tałanda et al., 2022*) provided by *Whiteside et al., 2024* in TNT v 1.5 (*Goloboff and Catalano, 2016*). We used *Whiteside et al., 2024* own data version, for example with (0) scored for character 1 and not including character 383. We recovered eight apomorphies at the squamate node, of which only three were recovered as unambiguous synapomorphies (boldface indicates the state as scored by *Whiteside et al., 2024* for †*Cryptovaranoides*):

> Ch. 138. Basisphenoid (or fused parabasisphenoid), ventral aspect, shape, concavity: single (0)/ divided (1)/ absent (**2**). 0 ->2
> Ch. 142. Prootics, alar crest: absent (0)/ present (**1**). 0->1

**Table 1.** Eight features described as synapomorphies for *Cryptovaranoides* + Squamata by *Whiteside et al., 2024*, the number for the character frolm *Tałanda et al. (2022)* versus the actual source for the character as claimed by *Whiteside et al. (2024)*, the actual clade-level synapomorphy for the character from the data they used, and its optimization.

Note: When listed as N/A, this means the character was not reconstructed as a synapomorphy of any clade and was not unambiguous in its transformation from one state to another. Only the first character in the table was found to be a synapomorphy, but was not unambiguous in its transformation from one state to another. Only the first character in the table was found to be a synapomorphy, but not of *Cryptovaranoides* + Squamata.

| Character | Number in Tałanda et al., 2022 | Study Cited in Whiteside et al., 2024 | Synapomorphy of | Unambiguous Optimization |
|---|---|---|---|---|
| Cephalic head of (mobile) quadrate with notch for the squamosal, peg-in-notch articulation with rod-shaped squamosal | 123 | *de Queiroz and Gauthier, 2020a* | Pan-Squamata | 1→0 |
| Vomer and maxilla meet at anterior margin of fenestra exochoanalis (see *Figure 6*) | 371 | *Gauthier et al., 2012* | N/A | N/A |
| Prominent choanal fossa on anterior margin of ventral surface of palatine | 100 | *Gauthier et al., 2012* | Lepidosauria | N/A |
| Subdivision of embryonic metotic fissure by the crista tuberalis into vagus (jugular) foramen and recessus scala tympani | 382 | *Simões et al., 2018*; *de Queiroz and Gauthier, 2020a* | N/A | N/A |
| No quadrate foramen | 118 | *Gauthier et al., 2012* | Lepidosauria | 1→0 |
| Medially positioned posterior mylohyoidal foramen on mandible (see *Figure 7*) | 163 | *Gauthier et al., 2012* | N/A | N/A |
| Fusion of exoccipitals and opisthotics forming an otoccipital | 151 | *Gauthier et al., 2012*; *de Queiroz and Gauthier, 2020a* | N/A | N/A |
| Trunk vertebrae lack intercentra | 237 | *de Queiroz and Gauthier, 2020a* | Pan-Unidentata | 1→0 |

Ch. 347. Prefrontal/palatine antorbital contact: absent (0) / narrow, forming less than 1/3 the transverse distance between the orbits (1) / contact broad, forming at least 1/2 the distance between the orbits (**2**). {01}->2

The eight features described by *Whiteside et al., 2024* (p. 19) as synapomorphies for †Cryptovaranoides +Squamata were also not inferred at their recovered crown squamate node from their own results. Rather, they were copied nearly verbatim from principally two sources *Gauthier et al., 2012*; *de Queiroz and Gauthier, 2020b* and presented as if they were recovered as synapomorphies by *Whiteside et al., 2024* (p. 19). In *Table 1*, the italicized text is from *Whiteside et al., 2024* (p.

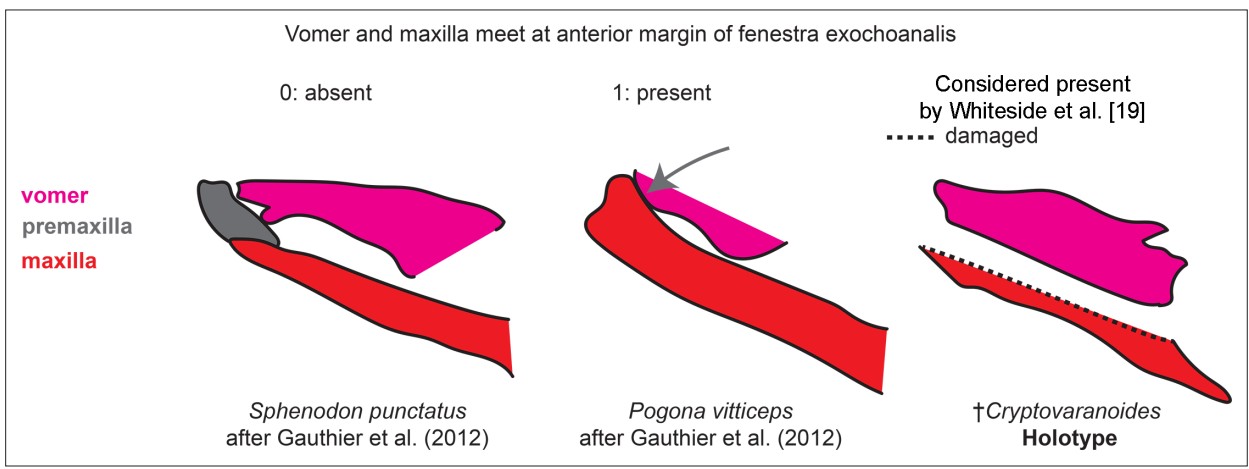

**Figure 6.** Anterior articulation of vomer and maxilla character scoring.

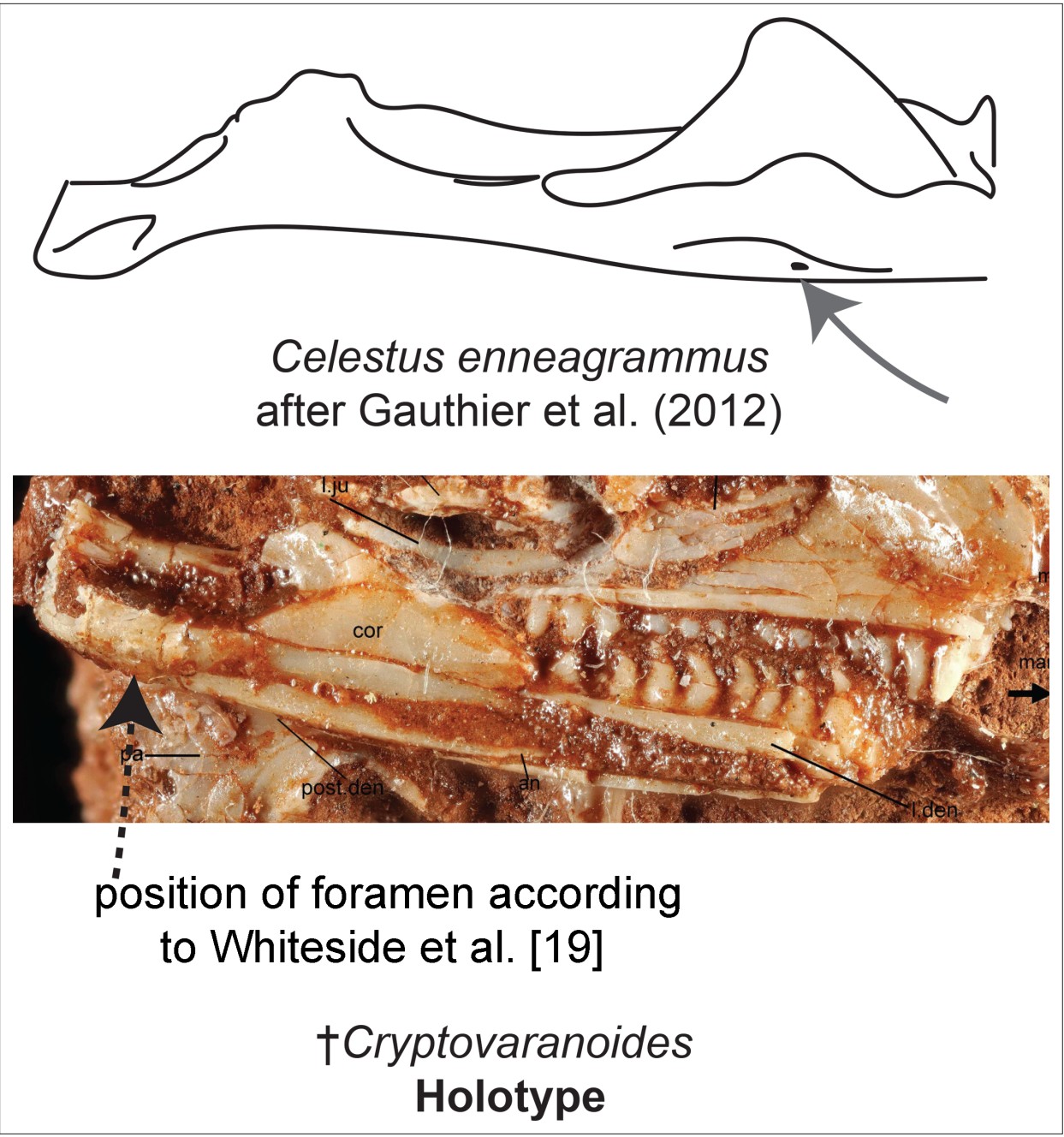

**Figure 7.** Presence of a medially positioned posterior mylohyoidal foramen on the mandible. As shown, there is no identifiable foramen on the mandible of †*Cryptovaranoides*. Bottom image of †*Cryptovaranoides* holotype skull is from ***Whiteside et al. (2024)*** Figure 2(b).

19), including their literature source for the supposed synapomorphy. The non-italicized text is the character number and state from ***Budd and Mann, 2024*** as recovered by us from the TNT analysis conducted (for this reassessment of ***Whiteside et al., 2024***). We also include where in our phylogeny the character state is recovered as synapomorphic (***Table 1***).

Our reanalysis shows that ***Whiteside et al., 2024*** did not diagnose †Cryptovaranoides +crown Squamata based on synapomorphies found by their own analyses as there were only three recovered in their results (see above). Instead, ***Whiteside et al., 2024*** provided ad hoc mischaracterizations of diagnostic features of crown Squamata from studies that did not include †*Cryptovaranoides* and then discussed and reviewed synapomorphies of Squamata from these sources (***Gauthier et al., 2012***; ***de Queiroz and Gauthier, 2020b***). Because each additional taxon has the possibility of inducing

character reoptimization, an empirical analysis of character optimization that includes a given new taxon is necessary to support referring said taxon to any particular clade. If †*Cryptovaranoides* was a crown squamate, it could plausibly show character distributions not previously sampled among the crown or stem group.

## Discussion

As we noted in *Brownstein et al., 2023*, the anatomy of †*Cryptovaranoides* is similar to many Late Triassic crown reptiles. For example, the quadrates of †*Cryptovaranoides* are closely comparable to those of archosauromorphs such as †*Prolacerta* (*Modesto and Sues, 2004*), †*Macrocnemus Miedema et al., 2020*, and †*Malerisaurus* (*Nesbitt et al., 2021*), with which †*Cryptovaranoides* shares the presence of a notch on the quadrate for an immobile articulation with the squamosal. *Whiteside et al., 2024* inference that the articulation with the squamosal was mobile, that is streptostylic as in squamates, is unfounded.

Similarly, the vertebrae and cervical ribs of the holotype of †*Cryptovaranoides* do not resemble those of squamates but are instead comparable to those of non-squamate neodiapsids, especially archosauromorphs (a point left unaddressed by *Whiteside et al., 2024*). For example, whereas fusion of the neural arches to the centra occurs during embryonic ossification in squamates (*Winchester and Bellairs, 1977*), in †*Cryptovaranoides,* unfused bony neural arches and centra are present, as commonly observed in archosaurs and other non-lepidosauromorph neodiapsids (*Brochu, 1996*; *Griffin et al., 2021*).

Several errors in *Whiteside et al., 2022* that we identified in *Brownstein et al., 2023* were challenged recently by *Whiteside et al., 2024*. The anatomical interpretations of *Brownstein et al., 2023* that are challenged in *Whiteside et al., 2024* are erroneously disputed in the latter study. *Whiteside et al., 2024* incorrectly interpreted neodiapsid and lepidosaur anatomy and consequently provided clearly erroneous scorings for †*Cryptovaranoides* for morphological character matrices. *Whiteside et al., 2024* also make several empirical errors, including the unjustified inflation of the †*Cryptovaranoides* hypodigm based on contradictory arguments and analytical problems (e.g. selecting apomorphic characters from the literature supporting their preferred placement of †*Cryptovaranoides* instead of using characters empirically obtained from phylogenetic ancestral-state reconstruction).

We end this comment with a perspective on the fossil record of neodiapsid evolution. The anatomies and morphologies that diagnose crown group squamates are many and varied, and, except for a few features, hardly universally distributed amongst the living and fossil members of the crown. They are themselves the product of some 250 million years of evolutionary time and would not have evolved in a linear fashion. Rather, phylogenetic analyses of diverse extinct and living reptile clades have shown that the squamate *bauplan* originated in the context of extensive mosaic and homoplastic osteological character evolution (*Simões et al., 2018*; *Brownstein et al., 2022*; *Simões et al., 2022a*; *Meyer et al., 2023*; *Jenkins et al., 2024*). We do not doubt that members of Pan-Squamata were present during the Triassic; this is supported by the fossil record (*Simões et al., 2018*) as well as numerous time-calibrated phylogenies based on genomic (*Title et al., 2024*; *Burbrink et al., 2020*), morphological (*Brownstein et al., 2022*; *Tałanda et al., 2022*; *Meyer et al., 2023*), and total-evidence (*Simões et al., 2018*; *Simões et al., 2022a*) data. However, *Whiteside et al., 2022* and *Whiteside et al., 2024* have provided no compelling osteological evidence that supports the presence of crown Squamata in the Triassic.

## Additional information

### Funding

| Funder | Grant reference number | Author |
| --- | --- | --- |
| Natural Sciences and Engineering Research Council of Canada | RGPIN-2022-03164 | Michael W Caldwell |

| Funder | Grant reference number | Author |
|---|---|---|

The funders had no role in study design, data collection and interpretation, or the decision to submit the work for publication.

## Author contributions

Michael W Caldwell, Conceptualization, Resources, Formal analysis, Investigation, Visualization, Methodology, Writing – original draft, Project administration, Writing – review and editing; Chase D Brownstein, Conceptualization, Formal analysis, Investigation, Methodology, Writing – original draft, Writing – review and editing; Dalton L Meyer, Simon G Scarpetta, Michael SY Lee, Investigation, Writing – review and editing; Tiago R Simões, Conceptualization, Investigation, Methodology, Writing – original draft, Writing – review and editing

## Author ORCIDs

Michael W Caldwell https://orcid.org/0000-0002-2377-3925
Tiago R Simões http://orcid.org/0000-0003-4716-649X

Reviewer #1 (Public review): https://doi.org/10.7554/eLife.107021.3.sa1
Reviewer #2 (Public review): https://doi.org/10.7554/eLife.107021.3.sa2
Reviewer #3 (Public review): https://doi.org/10.7554/eLife.107021.3.sa3
Author response https://doi.org/10.7554/eLife.107021.3.sa4

# Additional files

## Supplementary files
MDAR checklist

## Data availability
There are no new datasets generated for this study.

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
