## [Editor Report · eLife Assessment]

Cryptovaranoides, a Late Triassic animal (some 230 Ma old), was originally described as a possibly anguimorph squamate, i.e., more closely related to snakes and some extant lizards than to other extant lizards, making Squamata much older than previously thought and providing a new calibration date inside it. Following a rebuttal and a defense, this fourth **important** contribution to the debate makes a **convincing** argument that Cryptovaranoides is not a squamate. Further comparisons to potentially closely related animals such as early lepidosauromorphs would greatly benefit this study, and parts of the text require clarification.

---

## [Referee Report · Reviewer #1 (Public review)]

In the Late Triassic (around 230 Ma ago), southern Wales and adjacent parts of England were a karst landscape. The caves and crevices accumulated remains of small vertebrates. These fossil-rich fissure fills are being exposed in limestone quarrying. In 2022 (reference 13 of the article), a partial articulated skeleton and numerous isolated bones from one fissure fill were named Cryptovaranoides microlanius and described as the oldest known squamate - the oldest known animal, by some 50 Ma, that is more closely related to snakes and some extant lizards than to other extant lizards. This would have considerable consequences for our understanding of the evolution of squamates and their closest relatives, especially for its speed and absolute timing, and was supported in the same paper by phylogenetic analyses based on different datasets.

In 2023, the present authors published a rebuttal (ref. 18) to the 2022 paper, challenging anatomical interpretations and the irreproducible referral of some of the isolated bones to Cryptovaranoides. Modifying the datasets accordingly, they found Cryptovaranoides outside Squamata and presented evidence that it is far outside. In 2024 (ref. 19), the original authors defended most of their original interpretation and presented some new data, some of it from newly referred isolated bones. The present article discusses anatomical features and the referral of isolated bones in more detail, documents some clear misinterpretations, argues against the widespread but not justifiable practice of referring isolated bones to the same species as long as there is merely no known evidence to the contrary, further argues against comparing newly recognized fossils to lists of diagnostic characters from the literature as opposed to performing phylogenetic analyses and interpreting the results, and finds Cryptovaranoides outside Squamata again.

Although a few of the character discussions can probably still be improved, I see no sign that the discussion is going in circles or otherwise becoming unproductive. I can even imagine that the present contribution will end it.

---

## [Referee Report · Reviewer #2 (Public review)]

Congratulations on this revised manuscript on the phylogenetic affinities of Cryptovaranoides, and thank you for your modifications to this manuscript following review.

This manuscript offers a careful review of the features used to hypothesize the placement of Cryptovaranoides within crown Squamata and instead suggests that this taxon represents an earlier-diverging reptile. This work therefore reconciles morphological and molecular data regarding lizard origins, which is an important contribution to the field of vertebrate paleontology.

The authors have improved their manuscript following reviewer comments and now provide more thorough comparisons with other early reptiles and archosauromorphs, an improvement over early versions of this paper. Changes to these comparative descriptions provide important rationale concerning the absence of superficially squamate-like features in Cryptovaranoides.

The evolutionary relationships of Cryptovaranoides among reptiles will certainly be a matter of debate until detailed anatomical descriptions of this taxon and other putative lepidosauromorphs are published. However, it can now be said with confidence that the presence of any crown squamate in the Permian or Triassic is unlikely and should be met with skepticism, the same sort of skepticism provided in this manuscript.

---

## [Referee Report · Reviewer #3 (Public review)]

Summary:

The study provides an interesting contribution to our understanding of Cryptovaranoides relationships, which is a matter of intensive debate among researchers. The authors have modified the manuscript according to most of my suggestions. My main concerns are about the wording of some statements but the authors have the right to put it as they want in the end. Overall the discussion and data are well prepared. I would recommend to publish the manuscript after very minor revisions.

Strengths:

Detailed analysis of the discussed characters. Illustrations of some comparative materials.

Weaknesses:

Abstract: "Our team challenged this identification and instead suggested †Cryptovaranoides had unclear affinities to living reptiles"

Unfortunately I have to disagree again. "unclear affinities to living reptiles" can mean anything including a crown lizard. First, the 2023 paper clearly rejected the squamate hypothesis and presented some evidence that potentially places Cryptovaranoides among Archosauromorpha. In this context "unclear where it would belong within the latter" does not really matter. Second, we are not discussing here if Cryptovaranoides is a squamate or a stem-squamate. We have many more options on the table, so "unclear affinities" is too imprecise. Please change it to "could be an archosauromorph or an indeterminate neodiapsid" in the abstract to show the scale of conflicting evidence.

---

## [Author Response]

The following is the authors’ response to the original reviews.

**Reviewer #1 (Public review):**
In the Late Triassic and Early Jurassic (around 230 to 180 Ma ago), southern Wales and adjacent parts of England were a karst landscape. The caves and crevices accumulated remains of small vertebrates. These fossil-rich fissure fills are being exposed in limestone quarrying. In 2022 (reference 13 of the article), a partial articulated skeleton and numerous isolated bones from one fissure fill of end-Triassic age (just over 200 Ma) were named Cryptovaranoides microlanius and described as the oldest known squamate - the oldest known animal, by some 20 to 30 Ma, that is more closely related to snakes and some extant lizards than to other extant lizards. This would have considerable consequences for our understanding of the evolution of squamates and their closest relatives, especially for their speed and absolute timing, and was supported in the same paper by phylogenetic analyses based on different datasets.In 2023, the present authors published a rebuttal (reference 18) to the 2022 paper, challenging anatomical interpretations and the irreproducible referral of some of the isolated bones to Cryptovaranoides. Modifying the datasets accordingly, they found Cryptovaranoides outside Squamata and presented evidence that it is far outside. In 2024 (reference 19), the original authors defended most of their original interpretation and presented some new data, some of it from newly referred isolated bones. The present article discusses anatomical features and the referral of isolated bones in more detail, documents some clear misinterpretations, argues against the widespread but not justifiable practice of referring isolated bones to the same species as long as there is merely no known evidence to the contrary, further argues against comparing newly recognized fossils to lists of diagnostic characters from the literature as opposed to performing phylogenetic analyses and interpreting the results, and finds Cryptovaranoides outside Squamata again.Although a few of the character discussions and the discussion of at least one of the isolated bones can probably still be improved (and two characters are addressed twice), I see no sign that the discussion is going in circles or otherwise becoming unproductive. I can even imagine that the present contribution will end it.

We appreciate the positive response from reviewer 1!

**Reviewer #2 (Public review):**
Congratulations on this thorough manuscript on the phylogenetic affinities of Cryptovaranoides.

Thank you.

Recent interpretations of this taxon, and perhaps some others, have greatly changed the field's understanding of reptile origins- for better and (likely) for worse.

We agree, and note that while it is possible for challenges to be worse than the original interpretations, both the original and subsequent challenges are essential aspects of what make science, science.

This manuscript offers a careful review of the features used to place Cryptovaranoides within Squamata and adequately demonstrates that this interpretation is misguided, and therefore reconciles morphological and molecular data, which is an important contribution to the field of paleontology. The presence of any crown squamate in the Permian or Triassic should be met with skepticism, the same sort of skepticism provided in this manuscript.

We agree and add that every testable hypothesis requires skepticism and testing.

I have outlined some comments addressing some weaknesses that I believe will further elevate the scientific quality of the work. A brief, fresh read‑through to refine a few phrases, particularly where the discussion references Whiteside et al. could also give the paper an even more collegial tone.

We have followed Reviewer 2’s recommendations closely (see below) and have justified in our responses if we do not fully follow a particular recommendation.

This manuscript can be largely improved by additional discussion and figures, where applicable. When I first read this manuscript, I was a bit surprised at how little discussion there was concerning both non-lepidosauromorph lepidosaurs as well as stem-reptiles more broadly. This paper makes it extremely clear that Cryptovaranoides is not a squamate, but would greatly benefit in explaining why many of the characters either suggested by former studies to be squamate in nature or were optimized as such in phylogenetic analyses are rather widespread plesiomorphies present in crownward sauropsids such as millerettids, younginids, or tangasaurids. I suggest citing this work where applicable and building some of the discussion for a greatly improved manuscript. In sum:(1) The discussion of stem-reptiles should be improved. Nearly all of the supposed squamate features in Cryptovaranoides are present in various stem-reptile groups. I've noted a few, but this would be a fairly quick addition to this work. If this manuscript incorporates this advice, I believe arguments regarding the affinities of Cryptovaranoides (at least within Squamata) will be finished, and this manuscript will be better off for it.(2) I was also surprised at how little discussion there was here of putative stem-squamates or lepidosauromorphs more broadly. A few targeted comparisons could really benefit the manuscript. It is currently unclear as to why Cryptovaranoides could not be a stem-lepidosaur, although I know that the lepidosaur total-group in these manuscripts lacks character sampling due to their scarcity.

We are responding to (1) and (2) together. We agree with the Reviewer that a thorough comparison of Cryptovaranoides to non-lepidosaurian reptiles is critical. This is precisely what we did in our previous study: Brownstein et al. (2023)— see main text and supplementary information therein. As addressed therein, there is a substantial convergence between early lepidosaurs and some groups of archosauromorphs (our inferred position for Cryptovaranoides). Many of those points are not addressed in detail here in order to avoid redundancy and are simply referenced back to Brownstein et al. (2023). Secondly, stem reptiles (i.e., non-lepidosauromorphs and non-archosauromorphs), such as suggested above (millerettids, younginids, or tangasaurids), are substantially more distantly related to Cryptovaranoides (following any of the published hypotheses). As such, they share fewer traits (either symplesiomorphies or homoplasies), and so, in our opinion, we would risk directing losing the squamate-focus of our study.

We thus respectfully decline to engage the full scope of the problem in this contribution, but do note that this level of detailed work would make for an excellent student dissertation research program.

(3) This manuscript can be improved by additional figures, such as the slice data of the humerus. The poor quality of the scan data for Cryptovaranoides is stated during this paper several times, yet the scan data is often used as evidence for the presence or absence of often minute features without discussion, leaving doubts as to what condition is true. Otherwise, several sections can be rephrased to acknowledge uncertainty, and probably change some character scorings to '?' in other studies.

We strongly agree with the reviewer. Unfortunately, the original publication (Whiteside et al., 2021) did not make available the raw CT scan data to make this possible. As noted below in the Responses to Recommendations Section, we only have access to the mesh files for each segmented element. While one of us has observed the specimens personally, we have not had the opportunity to CT scan the specimens ourselves.

**Reviewer #3 (Public review):**
Summary:The study provides an interesting contribution to our understanding of Cryptovaranoides relationships, which is a matter of intensive debate among researchers. My main concerns are in regard to the wording of some statements, but generally, the discussion and data are well prepared. I would recommend moderate revisions.Strengths:(1) Detailed analysis of the discussed characters.(2) Illustrations of some comparative materials.

Thank you for noting the strengths inherent to our study.

Weaknesses:Some parts of the manuscript require clarification and rewording.One of the main points of criticism of Whiteside et al. is using characters for phylogenetic considerations that are not included in the phylogenetic analyses therein. The authors call it a "non-trivial substantive methodological flaw" (page 19, line 531). I would step down from such a statement for the reasons listed below:(1) Comparative anatomy is not about making phylogenetic analyses. Comparative anatomy is about comparing different taxa in search of characters that are unique and characters that are shared between taxa. This creates an opportunity to assess the level of similarity between the taxa and create preliminary hypotheses about homology. Therefore, comparative anatomy can provide some phylogenetic inferences.That does not mean that tests of congruence are not needed. Such comparisons are the first step that allows creating phylogenetic matrices for analysis, which is the next step of phylogenetic inference. That does not mean that all the papers with new morphological comparisons should end with a new or expanded phylogenetic matrix. Instead, such papers serve as a rationale for future papers that focus on building phylogenetic matrices.

We agree completely. We would also add that not every study presenting comparative anatomical work need be concluded with a phylogenetic analysis.

Our criticism of Whiteside et al. (2022) and (2024) is that these studies provided many unsubstantiated claims of having recovered synapomorphies between Cryptovaranoides and crown squamates without actually having done so through the standard empirical means (i.e., phylogenetic analysis and ancestral state reconstruction). Both Whiteside et al. (2022) and (2024) indicate characters presented as ‘shared with squamates’ along with 10 characters presented as synapomorphies (10). However, their actual phylogenetically recovered synapomorphies were few in number (only 3) and these were not discussed.

Furthermore, Whiteside et al. (2022) and (2024) comparative anatomy was restricted to comparing †Cryptovaranoides to crown squamates., based on the assumption that †Cryptovaranoides was a crown squamate and thus only needed to be compared to crown squamates.

In conclusion, we respectfully, we maintain such efforts are “non-trivial substantive methodological flaw(s)”.

(2) Phylogenetic matrices are never complete, both in terms of morphological disparity and taxonomic diversity. I don't know if it is even possible to have a complete one, but at least we can say that we are far from that. Criticising a work that did not include all the possibly relevant characters in the phylogenetic analysis is simply unfair. The authors should know that creating/expanding a phylogenetic matrix is a never-ending work, beyond the scope of any paper presenting a new fossil.

Respectfully, we did not criticize previous studies for including an incomplete phylogeny. Instead, we criticized the methodology behind the homology statements made in Whiteside et al. (2022) and Whiteside et al. (2024).

(3) Each additional taxon has the possibility of inducing a rethinking of characters. That includes new characters, new character states, character state reordering, etc. As I said above, it is usually beyond the scope of a paper with a new fossil to accommodate that into the phylogenetic matrix, as it requires not only scoring the newly described taxon but also many that are already scored. Since the digitalization of fossils is still rare, it requires a lot of collection visits that are costly in terms of time.

We agree on all points, but we are unsure of what the Reviewer is asking us to do relative to this study.

(4) If I were to search for a true flaw in the Whiteside et al. paper, I would check if there is a confirmation bias. The mentioned paper should not only search for characters that support Cryptovaranoides affinities with Anguimorpha but also characters that deny that. I am not sure if Whiteside et al. did such an exercise. Anyway, the test of congruence would not solve this issue because by adding only characters that support one hypothesis, we are biasing the results of such a test.

We would refer the Reviewer to their section (1) on comparative anatomy. As we and the Reviewer have pointed out, Whiteside et al. did not perform comparative anatomical statements outside of crown Squamata in their original study. More specifically, Whiteside et al. (2022, Fig. 8) presented a phylogeny where Cryptovaranoides formed a clade with Xenosaurus within the crown of Anguimorpha or what they termed “Anguiformes”, and made comparisons to the anatomies of the legless anguids, Pseudopus and Ophisaurus. Whiteside et al. (2024), abandoned “Anguiformes”, maintained comparisons to Pseudopus and emphasized affinities with Anguimorpha but almost all of their phylogenies as published, they do not recover a monophyletic Angumimorpha unless amphisbaenians and snakes are considered to be anguimorphans. Thus, we agree that confirmation bias was inherent in their studies.

To sum up, there is nothing wrong with proposing some hypotheses about character homology between different taxa that can be tested in future papers that will include a test of congruence. Lack of such a test makes the whole argumentation weaker in Whiteside et al., but not unacceptable, as the manuscript might suggest. My advice is to step down from such strong statements like "methodological flaw" and "empirical problems" and replace them with "limitations", which I think better describes the situation.

We agree with the first sentence in this paragraph – there is nothing wrong with proposing character homologies between different taxa based on comparative anatomical studies. However, that is not what Whiteside et al. (2022) and (2024) did. Instead, they claimed that an ad hoc comparison of Cryptovaranoides to crown Squamata confirmed that Cryptovaranoides is in fact a crown squamate and likely a member of Anguimorpha. Their study did not recognize limitations, but rather, concluded that their new taxon pushed the age of crown Squamata into the Triassic.

As noted by Reviewer 2, such a claim, and the ‘data’ upon which it is based, should be treated with skepticism. We have elected to apply strong skepticism and stringent tests of falsification to our critique.

**Reviewer #1 (Recommendations for the authors):**
(1) Lines 596-598 promise the following: "we provide a long[-]form review of these and other features in Cryptovaranoides that compare favorably with non-squamate reptiles in Supplementary Material." You have kindly informed me that all this material has been moved into the main text; please amend this passage.

This has been deleted.

(2) Comments on science41: I would rather say "an additional role".

This has been edited accordingly.

43: Reconstructing the tree entirely from extant organisms and adding fossils later is how Hennig imagined it, because he was an entomologist, and fossil insects are, on average,e extremely rare and usually very incomplete (showing a body outline and/or wing venation and little or nothing else). He was wrong, indeed wrong-headed. As a historical matter, phylogenetic hypotheses were routinely built on fossils by the mid-1860s, pretty much as soon as the paleontologists had finished reading On the Origin of Species, and this practice has never declined, let alone been interrupted. As a theoretical matter, including as many extinct taxa as possible in a phylogenetic analysis is desirable because it breaks up long branches (as most recently and dramatically shown by Mongiardino Koch & Parry 2020), and while some methods and some kinds of data are less susceptible to long-branch attraction and long-branch repulsion than others, none are immune; and while missing data (on average more common in fossils) can actively mislead parametric methods, this is not the case with parsimony, and even in Bayesian inference the problem is characters with missing data, not taxa with missing data. Some of you have, moreover, published tip-dated phylogenetic analyses. As a practical matter, molecular data are almost never available from fossils, so it is, of course, true that analyses which only use molecular data can almost never include fossils; but in the very rare exceptions, there is no reason to treat fossil evidence as an afterthought.

We agree and have changed “have become” to “is.”

49-50, 59: The ages of individual fissure fills can be determined by biostratigraphy; as far as I understand, all specimens ever referred to Cryptovaranoides [13, 19] come from a single fill that is "Rhaetian, probably late Rhaetian (equivalent of Cotham Member, Lilstock Formation)" [13: pp. 2, 15].

We appreciate this comment; the recent literature, however, suggests that variable ages are implied by the biostratigraphy at the English Fissure Fills, so we have chosen to keep this as is. Also note that several isolated bones were not recovered with the holotype but were discussed by Whiteside et al. (2024). The provenance of these bones was not clearly discussed in that paper.

59-60: Why "putative"? Just to express your disagreement? I would do that in a less misleading way, for example: "and found this taxon as a crown-group squamate (squamate hereafter) in their phylogenetic analyses." - plural because [19] presented four different analyses of two matrices just in the main paper.

We have removed this word.

121-124: The entepicondylar foramen is homologous all the way down the tree to Eusthenopteron and beyond. It has been lost a quite small number of times. The ectepicondylar foramen - i.e., the "supinator" (brachioradialis) process growing distally to meet the ectepicondyle, fusing with it and thereby enclosing the foramen - goes a bit beyond Neodiapsida and also occurs in a few other amniote clades (...as well as, funnily enough, Eusthenopteron in later ontogeny, but that's independent).

We agree. However, the important note here is that the features on the humerus of Cryptovaranoides are not comparable (differ in location and morphology) to the ent- and ectepondylar foramina in other reptiles, as we discuss at length. As such, we have kept this sentence as is.

153: Yes, but you [18] mistakenly wrote "strong anterior emargination of the maxillary nasal process, which is [...] a hallmark feature of archosauromorphs" in the main text (p. 14) - and you make the same mistake again here in lines 200-206! Also, the fact [19: Figure 2a-c] remains that Cryptovaranoides did not have an antorbital fenestra, let alone an antorbital fossa surrounding it (a fossa without a fenestra only occurs in some cases of secondary loss of the fenestra, e.g., in certain ornithischian dinosaurs). Unsurprisingly, therefore, Cryptovaranoides also does not have an orbital-as-opposed-to-nasal process on its maxilla [19: Figure 2a-c].

Line 243-249 (in original manuscript) deal with the emargination of maxillary nasal process (but this does not imply a full antorbital fenestra). We explicitly state that this feature alone "has limited utility" for supporting archosauromorph affinity.

158-173: The problem here is not that the capitellum is not preserved; from amniotes and "microsaurs" to lissamphibians and temnospondyls, capitella ossify late, and larger capitella attach to proportionately larger concave surfaces, so there is nothing wrong with "the cavity in which it sat clearly indicates a substantial condyle in life". Instead, the problem is a lack of quantification (...as has also been the case in the use of the exact same character in the debate on the origin of lissamphibians); your following sentence (lines 173-175) stands. The rest of the paragraph should be drastically shortened.

We appreciate this comment. We note that the ontogenetic variation of this feature is in part the issue with the interpretation provided by Whiteside et al. (2024). The issue is the lack of consistency on the morphology of the capitellum in that study. We are unclear on what the reviewer means by ‘quantification,’ as the character in question is binary.

250-252: It's not going to matter here, but in any different phylogenetic context, "sphenoid" would be confusing given the sphenethmoid, orbitosphenoid, pleurosphenoid, and laterosphenoid. I actually recommend "parabasisphenoid" as used in the literature on early amniotes (fusion of the dermal parasphenoid and the endochondral basisphenoid is standard for amniotes).

We have added "(=parabasisphenoid)" on first use but retain use of sphenoid because in the squamate and archosauromorph literature, sphenoid (or basisphenoid) is used more frequently.

314-315: Vomerine teeth are, of course, standard for sarcopterygians. Practically all extant amphibians have a vomerine toothrow, for example. A shagreen of denticles on the vomer is not as widespread but still reaches into the Devonian (Tulerpeton).

We agree, but vomerine teeth are rare in lepidosaurs and archosaurs and occur only in very recent clades e.g. anguids and one stem scincoid. Their presence in amphibians is not directly relevant to the phylogenetic placement of Cryptovaranoides among reptiles.

372: Fusion was not scored as present in [13], but as unknown (as "partial" uncertainty between states 0 and 1 [19:8]), and seemingly all three options were explored in [19].

We politely disagree with the reviewer; state 1 is scored in Whiteside et al. (2024).

377-383: Together with the partially fused NHMUK PV R37378 [13: Figure 4B, C; 19: 8], this is actually an argument that Cryptovaranoides is outside but close to Unidentata. The components of the astragalus fuse so early in extant amniotes that there is just a single ossification center in the already fused cartilage, but there are Carboniferous and Permian examples of astragali with sutures in the expected places; all of the animals in question (Diadectes, Hylonomus, captorhinids) seem to be close to but outside Amniota. (And yet, the astragalus has come undone in chamaeleons, indicating the components have not been lost.) - Also, if NHMUK PV R37378 doesn't belong to a squamate close to Unidentata, what does it belong to? Except in toothless beaks, premaxillary fusion is really rare; only molgin newts come to mind (and age, tooth size, and tooth number of NHMUK PV R37378 are wholly incompatible with a salamandrid).

The relevance of the astragalus is to the current discussion is unclear as we do not mention this element in our manuscript. We discuss the fusion in the premaxillae in response to previous comment.

471-474: That thing is concave. (The photo is good enough that you can enlarge it to 800% before it becomes too pixelated.) It could be a foramen filled with matrix; it does not look like a grain sticking to the outside of the bone. Also, spell out that you're talking about "suc.fo" in Figure 3j.

We are also a bit confused about this comment, as we state:

“Finally, we note here that Whiteside et al. [19] appear to have labeled a small piece of matrix attached to a coracoid that they refer to †C. microlanius as the supracoroacoid [sic] foramen in their figure 3, although this labeling is inferred because only “suc, supracoroacoid [sic]” is present in their figure 3 caption.” (L. 519-522, P. 17). We cannot verify that this structure is concave, as so we keep this text as is.

476-489: [19] conceded in their section 4.1 (pp. 11-12) that the atlas pleurocentrum, though fused to the dorsal surface of the axis intercentrum as usual for amniotes and diadectomorphs, was not fused to the axis pleurocentrum.

This is correct, as we note in the MS. The issue is whether these elements are clearly identifiable.

506-510: [19:12] did identify what they considered a possible ulnar patella, illustrated it (Figure 4d), scored it as unknown, and devoted the entire section 4.4 to it.512-523: What I find most striking is that Whiteside et al., having just discovered a new taxon, feel so certain that this is the last one and any further material from that fissure must be referable to one of the species now known from there.

We agree with these points and believe we have devoted adequate text to addressing them. Note that the reviewer does not recommend any revisions to these sections.

553: Not that it matters, but I'm surprised you didn't use TNT 1.6; it came out in 2023 and is free like all earlier versions.

We have kept this as is following the reviewer comment, and because we were interested in replicating the analyses in the previous publications that have contributed to the debate about the identity of this taxon. For the present simple analyses both versions should perform identically, as the search algorithms for discrete characters are identical across these versions.

562: Is "01" a typo, or do you mean "0 or 1"? In that case, rather write "0/1" or "{01}".

This has been corrected to {01}

(3) Comments on nomenclature and terminology55, 56: Delete both "...".

This has been corrected.

100: "ent- and ectepicondylar"

For clarity, we have kept the full words.

107-108: I understand that "high" is proximal and "low" is distal, but what is "the distal surface" if it is not the articular surface in the elbow joint?

This has been corrected.

120: "stem pan-lepidosaurs, and stem pan-squamates"; Lepidosauria and Squamata are crown groups that don't contain their stems

This has been corrected.

122, 123: Italics for Claudiosaurus and Delorhynchus.

This has been corrected.

130: Insert a space before "Tianyusaurus" (it's there in the original), and I recommend de-italicizing the two genus names to keep the contrast (as you did in line 162).

This has been corrected.

130, 131: Replace both "..." by "[...]", though you can just delete the second one.

This has been corrected.

174: Not a capitulum, but a grammatically even smaller (double diminutive) capitellum.

This has been corrected.

209, 224, Table 1: Both teams have consistently been doing this wrong. It's "recessus scalae tympani". The scala tympani ("ladder/staircase of the [ear]drum") isn't the recess, it's what the recess is for; therefore, the recess is named "recess of the scala tympani", and because there was no word for "of" in Classical Latin ("de" meant "off" and "about"), the genitive case was the only option. (For the same reason, the term contains "tympani", the genitive of "tympanum".)

This has been corrected.

415-425: This is a terminological nightmare. Ribs can have (and I'm not sure this is exhaustive): (a) two separate processes (capitulum, tuberculum) that each bear an articulating facet, and a notch in between; (b) the same, but with a non-articulating web of bone connecting the processes; (c) a single uninterrupted elongate (even angled) articulating facet that articulates with the sutured or fused dia- and parapophysis; (d) a single round articulating facet. Certainly, (a) is bicapitate and (d) is unicapitate, but for (b) and (c) all bets are off as to how any particular researcher is going to call them. This is a known source of chaos in phylogenetic analyses. I recommend writing a sentence or three on how the terms "unicapitate" & "bicapitate" lack fixed meanings and have caused confusion throughout tetrapod phylogenetics, and that the condition seen in Cryptovaranoides is nonetheless identical to that in archosauromorphs.

This has been added: “This confusion in part stems from the lack of a fixed meaning for uni- and bicapitate rib heads; in any case, †C. microlanius possesses a condition identical to archosauromorphs as we have shown.” (L.475-477, P.16).

439-440: Other than in archosaurs, some squamates and Mesosaurus, in which sauropsids are dorsal intercentra absent?

We are unclear about the relevance of the question to this section. The issue at hand is that some squamate lineages possess dorsal intercentra, so the absence of dorsal intercentra cannot be considered a squamate synapomorphy without the optimization of this feature along a phylogeny (which was not accomplished by Whiteside et al.).

458: prezygapophyses.

This has been corrected.

516: "[...]".

This has been corrected.

566: synapomorphies.

This has been corrected.

587: Macrocnemus.

This has been corrected.

585: I strongly recommend either taking off and nuking the name Reptilia from orbit (like Pisces) or using it the way it is defined in Phylonyms, namely as the crown group (a subset of Neodiapsida). Either would mean replacing "neodiapsid reptiles" with "neodiapsids".

This has been corrected to “neodiapsids.”

625: Replace "inclusive clades" by "included clades", "component clades", "subclades", or "parts," for example.

This has been kept as is because “inclusive clades” is common terminology and is used extensively in, for example, the PhyloCode.

659: Please update.

References are updated.

Fig. 8: Typo in Puercosuchus.

This has been corrected.

(4) Comments on style and spellingYou inconsistently use the past and the present tense to describe [13, 19], sometimes both in the same sentence (e.g., lines 323 vs. 325). I recommend speaking of published papers in the past tense to avoid ascribing past views and acts to people in their present state.

This has been corrected to be more consistent throughout the manuscript.

48: Remove the second comma.

This has been corrected.

91: Replace "[13] and WEA24" by "[13, 19]".

This has been corrected.

100: Commas on both sides of "in fact" or on neither

This has been corrected.

117: I recommend "the interpretation in [19]". I have nothing against the abbreviation "WEA24", but you haven't defined it, and it seems like a remnant of incomplete editing. - That said, eLife does not impose a format on such things. If you prefer, you can just bring citation by author & year back; in that case, this kind of abbreviation would make perfect sense (though it should still be explicitly defined).129, 145: Likewise.

We have modified this [13] and [19] where necessary.

192-198: Surely this should be made part of the paragraph in lines 158-175, which has the exact same headline?

This has been corrected.

200-206: Surely this should be made part of the paragraph in lines 148-156, which has the exact same headline?

These sections deal with different issues pertaining to the analyses of Whiteside et al. (2024) and so we have kept to organization as is.

214: Delete "that".

This has been deleted.

312: "Vomer" isn't an adjective; I'd write "main vomer body" or "vomer's main body" or "main body of the vomer".

This has been corrected.

350: "figured"

This has been corrected.

400: Rather, "rearticulated" or "worked to rearticulate"? - And why "several"? Just write "two". "Several" implies larger numbers.

These issues have been corrected.

448, 500: As which? As what kind of feature? I'm aware that "as such" is fairly widely used for "therefore", but it still confuses me every time, and I have to suspect I'm not the only one. I recommend "therefore" or "for this reason" if that is what you mean.

“As such” has been deleted.

452: Adobe Reader doesn't let me check, but I think you have two spaces after "of".

This has been corrected.

514, 539, 546, 552, 588, Fig. 3, 5, 6, Table 1: "WEA24" strikes again.

This has been corrected.

515: Remove the parentheses.

This has been corrected.

531: Insert a space after the period.

This has been corrected.

532: Remove both commas and the second "that".

This has been corrected.

538: Remove the comma.

This has been kept as is because changing it would render the sentence grammatically incorrect.

545: "[...]" or, better, nothing.

This has been corrected.

547: Spaces on both sides of the dash or on neither (as in line 553).

This has been corrected.

552: Rather, "conducted a parsimony analysis".

This has been corrected.

556: Space after "[19]".

This has been corrected.

560: Comma after "narrow".

This has been corrected.

600: Comma after "above" to match the one in the preceding line - there's an insertion in the sentence that must be flanked by commas on both sides.

This has been corrected.

603: Compound adjectives like "alpha-taxonomic" need a hyphen to avoid tripping readers up.

This has been corrected.

612: Similarly, "ancestral-state reconstruction" needs one to make immediately clear it isn't a state reconstruction that is ancestral but a reconstruction of ancestral states.

This has been corrected.

613: If you want to keep this comma, you need to match it with another after "Cryptovaranoides" in line 611.

We have kept this as is, because removing this comma would render the sentence grammatically incorrect.

615: Likewise, you need a comma after "and" because "except for a few features" is an insertion. The other comma is actually optional; it depends on how much emphasis you want to place on what comes after it.

this has been added.

622: Comma after "[48, 49]".

this has been added.

672: Missing italics and two missing spaces.

This has been corrected.

678, 680-681, 693, 700-701, 734, 742, 747, 788, 797, 799, 803, 808, 810-811, 814, 817, 820, 823, 828, 841, 843: Missing italics.

This has been corrected.

683, 689: These are book chapters. Cite them accordingly.

This has been corrected.

737: Missing DOI.

No DOI is available.

793: Missing Bolosaurus major; and I'd rather cite it as "2024" than "in press", and "online early" instead of "n/a".

This has been corrected.

835: Hoffstetter, RJ?

This has been corrected.

836: Is there something missing?

This has been corrected.

839: This is the same reference as number 20 (lines 683-684), and it is miscited in a different way...!

This has been corrected.

**Reviewer #2 (Recommendations for the authors):**
(1) There is a brief mention of a phylogenetic analysis being re-run, but it is unclear if any modifications (changes in scoring) based on the very observations were made. Please state this explicitly.

This is explained from lines 600-622, P.20-21, in the section “Apomorphic characters not empirically obtained.” "In order to check the characters listed by Whiteside et al. [19] (p.19) as “two diagnostic characters” and “eight synapomorphies” in support of a squamate identity for †Cryptovaranoides, we conducted a parsimony analysis of the revised version of the dataset [32] provided by Whiteside et al. [19] in TNT v 1.5 [91]. We used Whiteside et al.’s [19] own data version"

(2) Line 20: There is almost no discussion of non‑lepidosaur lepidosauromorphs. I suggest including this, as the archosauromorph‑like features reported in Cryptovaranoides appear rather plastic. Furthermore, diagnostic features of Archosauromorpha in other datasets (e.g., Ezcurra 2016 or the works of Spiekman) are notably absent (and unsampled) in Cryptovaranoides. Expanding this comparison would greatly strengthen the manuscript.

The brief discussion (although not absent) of non-lepidosaur lepidosauromorphs is largely a function of the poor fossil record of this grade. But where necessary, we do discuss these taxa. Also see our previous study (Brownstein et al. 2023) for an extensive discussion of characters relevant to archosauromorphs.

(3) Line 38: I suggest removing "Archosauromorpha" from the keywords. The authors make a compelling case that Cryptovaranoides is not a squamate, yet they do not fully test its placement within Archosauromorpha (as they acknowledge). Perhaps use "Reptilia" instead?

We have removed this keyword.

(4) Line 99: The authors' points here are well made and largely valid. The presence of the ent‑ and ectepicondylar foramina is indeed an amniote plesiomorphy and cannot confirm a squamate identity. Their absence, however, can be informative - although it is unclear whether the CT scans of the humerus are of sufficient resolution, and Figure 4 of Brownstein et al. looks hastily reconstructed (perhaps owing to limited resolution). Moreover, the foramina illustrated by Whiteside do resemble those of other reptiles, albeit possibly over‑prepared and exaggerated.

The issue with the noted figure is indeed due to poor resolution from the scans. Although we agree with the reviewer, we hesitate to talk about absence in this taxon being phylogenetically informative given the confounding influence of ontogeny.

(5) I encourage the authors to provide slice data to support the claim that the foramina are absent (which could certainly be correct!); otherwise, the assertion remains unsubstantiated.

We only have access to the mesh files of segmented bones, not the raw (reconstructed slice) data.

(6) PLEASE NOTE - because the specimen is juvenile, the apparent absence of the ectepicondylar foramen is equivocal: the supinator process develops through ontogeny and encloses this foramen (see Buffa et al. 2025 on Thadeosaurus, for example).

See above.

(7) Line 122: Italicize 'Delorhynchus'

This has been corrected.

(8) Lines 131‑132: I'd suggest deleting the final sentence; it feels a little condescending, and your argument is already persuasive.

This has been corrected.

(9) Line 129: Please note that owenettid "parareptiles" also lack this process, as do several other stem‑saurians. Its absence is therefore not diagnostic of Squamata.Also: Such plasticity is common outside the crown. Milleropsis and Younginidae develop this process during ontogeny, even though a lower temporal bar never fully forms.

We appreciate this point. See discussion later in the manuscript.

(11) Line 172: Consider adding ontogeny alongside taphonomy and preservation. A juvenile would likely have a poorly developed radial condyle, if any. Acknowledging this possibility will add some needed nuance.

This sentence has been modified, but we have not added in discussion of ontogeny here because it is not immediately relevant to refuting the argument about inference of the presence of this feature when it is not preserved.

(12) Line 177: The "septomaxilla" in Whiteside et al. (2024, Figure 1C) resembles the contralateral premaxilla in dorsal view, with the maxillary process on the left and the palatal (or vomerine) process on the right (the dorsal process appears eroded). The foramen looks like a prepalatal foramen, common to many stem and crown reptiles. Consequently, scoring the septomaxilla as absent may be premature; this bone often ossifies late. In my experience with stem‑reptile aggregations, only one of several articulated individuals may ossify this element.

We agree that presence of a late-ossifying septomaxilla cannot be ruled out, but our point remains (and in agreement with Referee) that scoring the septomaxilla as present based on the amorphous fragments is premature.

(13) Line 200: Tomography data should be shown before citing it. The posterior margin of the maxilla appears rather straight, and the maxilla itself is tall for an archosauromorph. It would be more convincing to score this feature as present only after illustrating the relevant slices - and, as you note, the trait is widespread among non‑archosauromorphs.

See above and Brownstein et al. (2023).

(14) Line 208: Well argued: how could Whiteside et al. confidently assign a disarticulated element? Their "vagus" foramen actually resembles a standard hypoglossal foramen - identical to that seen in many stem reptiles, which often have one large and one small opening.

Thank you!

(15) Line 248: Again, please illustrate this region. One cannot argue for absence without showing the slice data. Note that millerettids and procolophonians - contemporaneous with Cryptovaranoides - possess an enclosed vidian canal, so the feature is broadly distributed.

See above.

(16) Line 258: The choanal fossa is intriguing: originally created for squamate matrices, yet present (to varying degrees) in nearly every reptile I have examined. It is strongly developed in millerettids (see Jenkins et al. 2025 on Milleropsis and Milleretta) and younginids, much like in squamates - Tiago appropriately scores it as present. Thus, it may be more of a "Neodiapsida + millerettids" character. In any case, the feature likely forms an ordered cline rather than a simple binary state.

We agree and look forward to future study of this feature.

(17) Line 283: Bolosaurids are not diapsids and, per Simões, myself, and others, "Diapsida" is probably invalid, at least how it is used here. Better to say "neodiapsids" for choristoderes and "stem‑reptiles" or "sauropsids" for bolosaurids. Jenkins et al.'s placement is largely a function of misidentifying the bolosaurid stapes as the opisthotic.

We are not entirely clear on this point since bolosaurids are not mentioned in this section.

(18) Line 298: Here, you note that the CT scans are rather coarse, which makes some earlier statements about absence/presence less certain (e.g., humeral foramina). It may strengthen the paper to make fewer definitive claims where resolution limits interpretation.

We appreciate this point. However, in the case of the humeral foramina the coarseness of the scans is one reason why we question Whiteside et al. scoring of the presence of these features.

(19) Line 314: Multiple rows of vomerine teeth are standard for amniotes; lepidosauromorphs such as Paliguana and Megachirella also exhibit them (though they may not have been segmented in the latter's description). Only a few groups (e.g., varanopids, some millerettids) have a single medial row.

We appreciate this point and have added in those citations into the following added sentence: “Multiple rows of vomerine teeth are common in reptiles outside of Squamata [76]; the presence of only one row is restricted to a handful of clades, including millerettids [77,78], †Tanystropheus [49], and some [79], but not all [71,80] choristoderes.” (L. 360-363, P. 12).

(20) Line 317: This is likely a reptile plesiomorphy - present in all millerettids (e.g., Milleropsis and Milleretta per Jenkins et al.). Citing these examples would clarify that it is not uniquely squamate. Could it be secondarily lost in archosauromorphs?

We appreciate this point and have cited Jenkins et al. here. It is out of the scope of this discussion to discuss the polarity of this feature relative to Archosauromorpha.

(21) Line 336: Unfortunately, a distinct quadratojugal facet is usually absent in Neodiapsids and millerettids; where present, the quadratojugal is reduced and simply overlaps the quadrate.

We appreciate this point but feel that reviewing the distribution of this feature across all reptiles is not relevant to the text noted.

(22) Line 357: Pterygoid‑quadrate overlap is likely a tetrapod plesiomorphy. Whiteside et al. do not define its functional or phylogenetic significance, and the overlap length is highly variable even among sister taxa.

We agree, but in any case this feature is impossible to assess in Cryptovaranoides.

(23) Line 365: Another well‑written section - clear and persuasive.

Thank you!

(24) Line 385: The cephalic condyle is widespread among neodiapsids, so it is not uniquely squamate.

We agree.

(25) Character 391: Note that the frontal underlapping the parietal is widespread, appearing in both millerettids and neodiapsids such as Youngina.

We appreciate this point, but the point here deals with the fact that this feature is not observable in the holotype of Cryptovaranoides.

(26) Line 415: The "anterior process" is actually common among crown reptiles, including sauropterygians, so it cannot by itself place Cryptovaranoides within Archosauromorpha.

We agree but also note that we do not claim this feature unambiguously unites Cryptovaranoides with Archosauromorpha.

(28) Line 460: Yes - Whiteside et al. appear to have relabeled the standard amniote coracoid foramen. Excellent discussion.

Thank you!

(29) Line 496: While mirroring Whiteside's structure, discussing this mandibular character earlier, before the postcrania, might aid readability.

We have chosen to keep this structure as is.

(30) Lines 486-588: This section oversimplifies the quadrate articulation.

We are unclear how this is an oversimplification.

(31) Both Prolacerta and Macrocnemus possess a cephalic condyle and some mobility (though less than many squamates). In Prolacerta (Miedema et al. 2020, Figure 4), the squamosal posteroventral process loosely overlaps the quadrate head.

We assume this comment refers to the section "Peg-in-notch articulation of quadrate head"; we appreciate clarification that this feature occurs in variable extent outside squamates, but this does not affect our statement that the material of Cryptovaranoides is too poorly preserved to confirm its presence.

(32) Where is this process in Cryptovaranoides? It is not evident in Whiteside's segmentation of the slender squamosal - please illustrate.

We are unclear as to which section this comment refers.

(33) Additionally, the quadrate "conch" of Cryptovaranoides is well developed, bearing lateral and medial tympanic crests; the lateral crest is absent in the cited archosauromorphs.

We note that no vertebrate has a medial tympanic crest (it is always laterally placed for the tympanic membrane, when present). If this is what the reviewer refers to, this is a feature commonly found across all tetrapods bearing a tympanum attached to the quadrate (e.g., most reptiles), and so it is not very relevant phylogenetically. Regarding its presence in Cryptovaranoides, the lateral margin of the quadrate is broken (Brownstein et al., 2023), so it cannot be determined. This incomplete preservation also makes an interpretation of a quadrate conch very hard to determine. But as currently preserved, there is no evidence whatsoever for this feature.

(34) Line 591: The cervical vertebrae of Cryptovaranoides are not archosauromorph‑like. Archosauromorph cervicals are elongate, parallelogram‑shaped, and carry long cervical ribs-none of which apply here. As the manuscript lacks a phylogenetic analysis, including these features seems unnecessary. Should they be added to other datasets, I suspect Cryptovaranoides would align along the lepidosaur stem (though that remains to be tested).

We politely disagree. The reviewer here mentions that the cervical vertebrae of archosauromorphs are generally shaped differently from those in Cryptovaranoides. The description provided (“elongate, parallelogram‑shaped, and carry long cervical ribs-none”) is basically limited to protorosaurians (e.g., tanystropheids, Macrocnemus) and early archosauriforms. We note that archosauromorph cervicals are notoriously variable in shape, especially in the crown, but also among early archosauromorphs. Further, the cervical ribs, are notoriously similar among early archosauromorphs (including protorosaurians) and Cryptovaranoides, as discussed and illustrated in Brownstein et al., 2023 (Figs. 2 and 3), especially concerning the presence of the anterior process.

Further, we do include a phylogenetic analysis of the matrix provided in Whiteside et al. (2024) as noted in our results section. In any case, we direct the reviewer to our previous study (Brownstein et al., 2023), in which we conduct phylogenetic analyses that included characters relevant to this note.

**Reviewer #3 (Recommendations for the authors):**
(1) The authors should use specimen numbers all over the text because we are talking about multiple individuals, and the authors contest the previous affinity of some of them. For example, on page 16, line 447, they mention an isolated vertebra but without any number. The specimen can be identified in the referenced article, but it would be much easier for the reader if the number were also provided here

Agreed and added.

(2) Abstract: "Our team questioned this identification and instead suggested Cryptovaranoides had unclear affinities to living reptiles."That is very imprecise. The team suggested that it could be an archosauromorph or an indeterminate neodiapsid. Please change accordingly.

We politely disagree. We stated in our 2023 study that whereas our phylogenetic analyses place this taxon in Archosauromorpha, it remains unclear where it would belong within the latter. This is compatible with “unclear affinities to living reptiles”.

(3) Page 7, line 172: "Taphonomy and poor preservation cannot be used to infer the presence of an anatomical feature that is absent." Unfortunate wording. Taphonomy always has to be used to infer the presence or absence of anatomical features. Sometimes the feature is not preserved, but it leaves imprints/chemical traces or other taphonomic indicators that it was present in the organism. Please remove or rewrite the sentence.

We agree and have modified the sentence to read: “Taphonomy and poor preservation cannot be used alone to justify the inference that an anatomical feature was present when it is not preserved and there is no evidence of postmortem damage. In a situation when the absence of a feature is potentially ascribable to preservation, its presence should be considered ambiguous.” (L. 141-145, P.5).

(4) Page 4, line 91, please explain "WEA24" here, though it is unclear why this abbreviation is used instead of citation in the manuscript.

This has been corrected to Whiteside et al. [19].

(5) Page 6, line 144: "Together, these observations suggest that the presence of a jugal posterior process was incorrectly scored in the datasets used by WEA24 (type (ii) error)." That sentence is unclear. Why did the authors use "suggest"? Does it mean that they did not have access to the original data matrix to check it? If so, it should be clearly stated at the beginning of the manuscript.

See earlier; this has been modified and “suggest” has been removed.

(6) Page 7, line 174: "Finally, even in the case of the isolated humerus with a preserved capitulum, the condyle illustrated by Whiteside et al. [19] is fairly small compared to even the earliest known pan-squamates, such as Megachirella wachtleri (Figure 4)." Figure 4 does not show any humeri. Please correct.

The reference to figure 4 has been removed.

(7) Page 8, line 195-198: "This is not the condition specified in either of the morphological character sets that they cite [18,38], the presence of a distinct condyle that is expanded and is by their own description not homologous to the condition in other squamates." This is a bit unclear. Could the authors explain it a little bit further? How is the condition that is specified in the referred papers different compared to the Whiteside et al. description?

We appreciate this comment and have broken this sentence up into three sentences to clarify what we mean:

“The projection of the radial condyle above the adjacent region of the distal anterior extremity is not the condition specified in either of the morphological character sets that Whiteside et al. [19] cite [18,32]. The condition specified in those studies is the presence of a distinct condyle that is expanded. The feature described in Whiteside et al. [19] does not correspond to the character scored in the phylogenetic datasets.” (L.220-225, P.8).

(8) Page 16, line 446: "they observed in isolated vertebrae that they again refer to C. microlanius without justification". That is not true. The referred paper explains the attribution of these vertebrae to Cryptovaranoides (see section 5.3 therein). The authors do not have to agree with that justification, but they cannot claim that no justification was made. Please correct it here and throughout the text.

We have modified this sentence but note that the justification in Whiteside et al. (2024) lacked rigor. Whiteside et al. (2024) state: “Brownstein et al. [5] contested the affinities of three vertebrae, cervical vertebra NHMUK PV R37276, dorsal vertebra NHMUK PV R37277 and sacral vertebra NHMUK PV R37275. While all three are amphicoelous and not notochordal, the first two can be directly compared to the holotype. Cervical vertebra NHMUK PV R37276 is of the same form as the holotype CV3 with matching neural spine, ventral keel (=crest) and the posterior lateral ridges or lamina (figure 3c,d) shown by Brownstein et al. [5, fig. 1a]. The difference is that NHMUK PV R37276 has a fused neural arch to the pleurocentrum and a synapophysis rather than separate diapophysis and parapophysis of the juvenile holotype (figure 3c). Neurocentral fusion of the neural arch and centrum can occur late in modern squamates, ‘up to 82% of the species maximum size’ [28].

The dorsal surface of dorsal vertebra NHMUK PV R37277 (figure 3e) can be matched to the mid-dorsal vertebra in the †Cryptovaranoides holotype (figure 4d, dor.ve) and has the same morphology of wide, dorsally and outwardly directed, prezygapophyses, downwardly directed postzygapophyses and similar neural spine. It is also of similar proportions to the holotype when viewed dorsally (figures 3e and 4d), both being about 1.2 times longer anteroposteriorly than they are wide, measured across the posterior margin. The image in figure 4d demonstrates that the posterior vertebrae are part of the same spinal column as the truncated proximal region but the spinal column between the two parts is missing, probably lost in quarrying or fossil collection.”

This justification is based on pointing out the presence of supposed shared features between these isolated vertebrae and those in the holotype of Cryptovaranoides, even though none of these features are diagnostic for that taxon. We have changed the sentence in our manuscript to read:

“Whiteside et al. [19] concur with Brownstein et al. [18] that the diapophyses and parapophyses are unfused in the anterior dorsals of the holotype of †Cryptovaranoides microlanius, and restate that fusion of these structures is based on the condition they observed in isolated vertebrae that they refer to †C. microlanius based on general morphological similarity and without reference to diagnostic characters of †C. microlanius” (L. 502-507, P. 17).

(9) Figure 2. The figure caption lacks some explanations. Please provide information about affinity (e.g., squamate/gekkotan), ag,e and locality of the taxa presented. Are these left or right palatines? The second one seems to be incomplete, and maybe it is worth replacing it with something else?

The figure caption has been modified:

“Figure 2. Comparison of palatine morphologies. Blue shading indicates choanal fossa. Top image of †Cryptovaranoides referred left palatine is from Whiteside et al. [19]. Middle is the left palatine of †Helioscopos dickersonae (Squamata: Pan-Gekkota) from the Late Jurassic Morrison Formation [62]. Bottom is the right palatine of †Eoscincus ornatus (Squamata: Pan-Scincoidea) from the Late Jurassic Morrison Formation [31].”

(10) Figure 8. The abbreviations are not explained in the figure caption.

These have been added.